# Drag-and-Drop LLMs: Zero-Shot Prompt-to-Weights

Zhiyuan Liang[1][*]   Dongwen Tang[1]   Yuhao Zhou[1]   Xuanlei Zhao[1]   Mingjia Shi[1]
Wangbo Zhao[1]   Zekai Li[1]   Peihao Wang[2]   Konstantin Schürholt[3]   Damian Borth[3]
Michael M. Bronstein[4]   Yang You[1]   Zhangyang Wang[2][*]   Kai Wang[1][*]

[1]National University of Singapore, [2]UT Austin, [3]University of St. Gallen, [4]Oxford University

## Abstract

Modern Parameter-Efficient Fine-Tuning (PEFT) methods such as low-rank adaptation (LoRA) reduce the cost of customizing large language models (LLMs), yet still require a separate optimization run for every downstream dataset. We introduce **Drag-and-Drop LLMs (*DnD*)**, a prompt-conditioned parameter generator that eliminates per-task training by mapping a handful of unlabeled task prompts directly to LoRA weight updates. A lightweight text encoder distills each prompt batch into condition embeddings, which are then transformed by a cascaded hyper-convolutional decoder into the full set of LoRA matrices. Once trained in a diverse collection of prompt-checkpoint pairs, DnD produces task-specific parameters in seconds, yielding i) up to **12,000**$\times$ lower overhead than full fine-tuning, ii) average gains up to **30%** in performance over the strongest training LoRAs on unseen common-sense reasoning, math, coding, and multimodal benchmarks, and iii) robust cross-domain generalization despite never seeing the target data or labels. Our results demonstrate that prompt-conditioned parameter generation is a viable alternative to gradient-based adaptation for rapidly specializing LLMs. Our project is available at https://jerryliang24.github.io/DnD.

## 1 Introduction

Large Language Models (LLMs) such as GPT-4, Llama 2/3, Qwen2.5, and DeepSeek have rapidly become the backbone of contemporary natural-language processing and artificial intelligence more broadly, thanks to their internet-scale pre-training and transformer architectures [1, 19, 63, 36]. This pre-training endows a single model with broad *zero-shot* competence across mathematics, coding, reasoning, and even multimodal understanding [24, 12, 62, 29]. Yet, real-world deployments rarely stop at zero-shot use; they demand *task-specific* behavior that reflects internal data, domain jargon, or bespoke response styles. Parameter-Efficient Fine-Tuning (PEFT) aims to satisfy this demand by inserting a small set of trainable weights, most prominently the low-rank adapters of LoRA [25]. While LoRA allows to maintain the number of trainable parameters and storage overhead small by keeping the model frozen, the *wall-clock cost* remains very high: for example, adapting the lightest 0.5 B-parameter Qwen2.5 with LoRA still occupies four A100 GPUs for half a day [63]. Moreover, each downstream user or dataset requires its own optimization run, which quickly becomes the computational bottleneck for practitioners when deploying PEFT at massive scales.

We observe that a LoRA adapter is nothing more than a *function of its training data*: gradient descent "drags" the base weights towards a task-specific optimum (Figure 1). If that mapping from *prompts* to *weights* can be learned *directly*, we could bypass gradient descent altogether. Early work on *parameter generation* has shown that hyper-networks can synthesize billions of weights in minutes [51, 43, 59, 49, 53]. Yet they either ignore task conditioning or use simple binary embeddings.

---

[*]Zhiyuan, Zhangyang, and Kai are core contributors.

39th Conference on Neural Information Processing Systems (NeurIPS 2025).

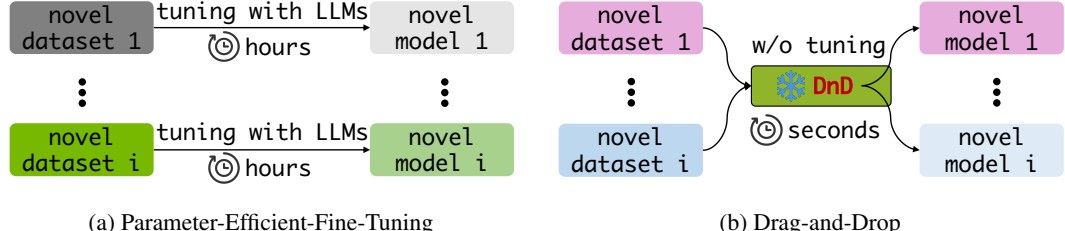

(a) Parameter-Efficient-Fine-Tuning     (b) Drag-and-Drop

Figure 1: *Left*: Parameter-efficient methods such as LoRA need *hours* to optimize LLMs in order to adapt them to novel datasets. *Right*: Our method adapts LLMs by directly generating LoRA matrices for novel datasets in *seconds* **without any tuning**.

Recent progress makes this goal attainable. RPG [58] is one of the first approaches to condition on task information and generate an entire classifier in a single pass, matching from-scratch training on previously unseen image classes in zero-shot. Translating that success to language, however, raises new obstacles. First, linguistic prompts carry orders of magnitude more semantic variation than the binary embeddings used by RPG. A practical generator must therefore ingest *rich task descriptors* and preserve their nuances. Second, an LLM in production may face hundreds of heterogeneous workloads, so the conditioning mechanism must scale gracefully while injecting task-specific cues with high fidelity. These challenges bring the need for a *compact yet expressive* representation that both captures salient features of the input texts and steers the hyper-network towards the corresponding region of LoRA weight space. Designing such a representation is the central challenge that our method, introduced next, is built to address.

We introduce **Drag-and-Drop LLMs (DnD)**, a prompt-conditioned hyper-generator that converts a handful of unlabeled task prompts into a complete set of LoRA adapters in seconds, eliminating any per-task optimization. DnD employs an off-the-shelf, lightweight text encoder to compress a given batch of prompts into conditional embeddings, which a cascaded hyper-convolutional decoder then expands into LoRA updates for every transformer layer.

On common-sense reasoning, mathematics, code-generation, and multimodal benchmarks, DnD cuts adaptation overhead by up to *12,000×* while yielding up to *30%* increased performance on *unseen* datasets compared with the strongest training LoRAs, and transfers seamlessly from 0.5B to 7B parameter backbones. By collapsing the classical "data→gradients→weights" loop into a single forward step, DnD challenges the notion that gradient descent is indispensable for model specialization and opens a new path where weights themselves become a new data modality and generative target conditioned on concise task descriptors.

Our main contributions are outlined as follows:

**New LLM adaptation paradigm.** We cast LoRA adaptation as direct generation of task-specific weights from raw prompts in a *novel* dataset and realize this mapping with a scalable hyper-generator, which is way more efficient than traditional tuning.

**Practical Architecture.** A frozen text encoder coupled with a hyper-convolutional decoder is able to generate large scale parameters while reducing adaptation overhead by four orders of magnitude.

**Comprehensive evaluation.** Experiments across reasoning, math, coding, and multimodal show up to 30% zero-shot gains on unseen datasets and smooth transfer across model sizes, highlighting DnD's impressive efficiency and versatility.

## 2 Drag-and-Drop Your LLMs

### 2.1 Preliminary

**Parameter-Efficient Fine-Tuning.** Parameter-Efficient Fine-Tuning (PEFT) saves training costs via introducing and tuning only a small number of additional parameters while keeping the original model weights frozen. This approach has been applied to LLMs and other foundation models, particularly in Low-Rank Adaptation (LoRA) manner, such as LLaMA [19] and Stable Diffusion [48]. The optimization process can be formulated as:

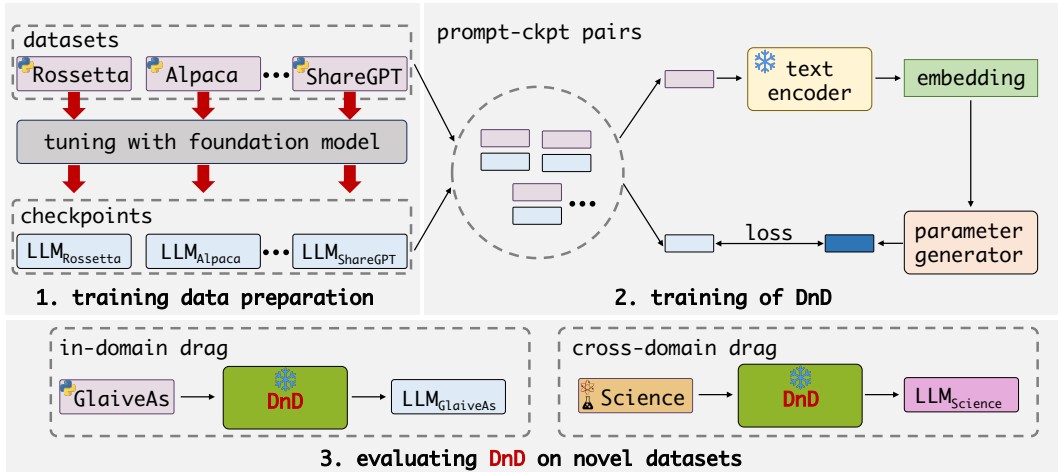

Figure 2: Our approach obtains dragging ability via two processes: prepare training data (*upper left*) and training the parameter generator (*upper right*). When preparing training data, we explicitly pair parameters with dataset-specific conditions. During training, DnD takes condition as input and generate parameters, using original parameters as supervision.

$$\min_{A,B} \mathcal{L}(W_0 + BA, \mathcal{D}), \tag{1}$$

where $W_0$ is the frozen original weight matrix, low-rank matrices $B \in \mathbb{R}^{d \times r}$ and $A \in \mathbb{R}^{r \times k}$ with $r \ll \min(d, k)$ are the only trainable parameters and $\mathcal{D}$ represents the fine-tuning dataset. Based on Equation 1, we can conclude that LoRA uses data $\mathcal{D}$ as raw material and optimization as the driving force to obtain weight space shift $\Delta W = BA$, thus making $\Delta W$ and $\mathcal{D}$ strongly associated.

**Parameter generation.** This approach treats models or trained weights as data, aiming to synthesize high-performing neural network parameters without conventional training. Recent advances, such as COND P-DIFF [27], RPG [58], SANE [53], and ORAL [28] have achieved controllable generation by incorporating conditioning mechanisms, allowing primitive personalized generation of parameters for simple datasets. The parameter generation process shares fundamental commonalities with PEFT, where conditions serve as raw materials and the parameter generator provides the driving force to produce target weights with specific properties.

One questions remains: Can we utilize parameter generation to effectively "drag-and-drop" LLMs' weights towards configurations better suited for a given novel task? By "drag-and-drop", we draw an analogy to our simple, *tuning-free* process that directly generates *task-specific weights*, associating it to the intuitive action of dragging a file and dropping it into place without further configuration.

**Key challenges.** Before addressing the above question, we analyze the potential challenges.

**Challenge 1:** How to equip the parameter generator with effective "drag-and-drop" ability? The generator should produce parameters that can effectively adapt LLMs towards specific tasks.

**Challenge 2:** How to enable adaptation without task-specific training? Traditional PEFT methods typically require training on new tasks, but can we achieve comparable performance by directly generating parameters without any fine-tuning on the target task?

**Challenge 3:** How to make the drag-and-drop function user-friendly and accessible? The generation mechanism should be simple and intuitive, enabling broader adoption and practical deployment.

## 2.2 Overview of DnD

To address the identified challenges, we present DnD as illustrated in Figure 2. As preparation, we first train and save LoRA adapters on various datasets. To develop the "drag-and-drop" capability, our approach should be aware of parameters' correlations with datasets. Consequently, we randomly pair collected checkpoints with prompt batches from their training data. A pre-trained text encoder then extracts embeddings from the prompts and feed them to our parameter generator. The generator features a simple architecture of cascaded pure convolutional decoder blocks (details in Section 2.5 and Appendix A.3). We optimize this generator using mean squared error (MSE) loss between the generated and original tokenized model weights. During inference, we evaluate our approach in both

in-domain and cross-domain scenarios: simply feeding prompts from novel datasets (not seen during training) to DnD to obtain tailored model parameters with one single forward pass.

## 2.3 Data Preparation of DnD.

**Checkpoint collection.** We collect the checkpoints across various datasets, *i.e.*, serving as diverse supervisions, to equip the capability of DnD. The collection process follows previous parameter generation works [59, 58]: training for specified epochs, then performing iterative fine-tuning while preserving checkpoints at each iteration (more details in Appendix A.1).

**The role of prompts.** Recent studies [38, 66] demonstrate that samples from different datasets exhibit distinct features, *i.e.*, samples could be considered as "fingerprints" of specified datasets (tasks). Based on this observation, we utilize data samples (prompts) as representative proxies for their respective datasets (tasks). To establish data-parameter mapping, we incorporate prompts from the datasets used to train these checkpoints. These prompts contain dataset-specific features, enabling the generator to infer appropriate "dragging" directions for models across various tasks.

**Prompt-checkpoint pairing.** Based on the above analysis, the next important question is: how to utilize these elements to equip DnD with "drag-and-drop" ability in training? Given a dataset $P$, we first divide it into non-overlapping prompt batches $[p_1, \cdots, p_i, \cdots, p_I]$. We note the trained LLM checkpoints of this dataset as $M = [m_1, \cdots, m_j, \cdots, m_J]$. We randomly pick a batch of prompts and a corresponding checkpoint. The process can be formulated as,

$$[p_1, \cdots, p_i, \cdots, p_I] \xrightarrow{\text{randomly pick}} \{p_i, m_j\} \xleftarrow{\text{randomly pick}} [m_1, \cdots, m_j, \cdots, m_J], \qquad (2)$$

where $\{p_i, m_j\}$ serves as a pair for parameter generator training. Prompt $p_i$ and checkpoint $m_j$ serve as the input and supervision, respectively.

## 2.4 Prompt Embedding.

For each batch of prompts, we employ a open-sourced text encoder to extract embeddings that serve as inputs for the parameter generator. This extraction process can be formally represented as:

$$c_i = \text{Encoder}(p_i, \theta), \qquad (3)$$

where $\text{Encoder}(\cdot, \cdot)$ denotes the embedding extraction function parameterized by $\theta$, and $c_i$ represents the extracted embedding corresponding to prompt $p_i$. By default, we leverage an encoder-based language model architecture [14] for prompt embedding. In the experimental section, we further explore and quantitatively evaluate alternative embedding approaches, including word2vec representations [44], encoder-decoder architecture [46], and decoder-only language models [63].

## 2.5 Training and Inference of DnD.

Figure 3: Block details of parameter generator. Each block of hyper-convolution contains three hyper-convolution modules, extracting and fusing features in different dimensions. More details are in Appendix A.3.

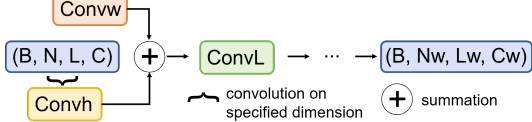

**Structure of parameter generator.** Different from diffusion-based parameter generation [59, 27, 58], we use a hyper-convolutional decoder to learn the mapping between input prompts and parameters. That design mainly considers efficiency, as the decoder-only structure has shown its superiority in LLMs [1, 19, 63, 22]. We show the block details of the decoder in the right part (Figure 3). We assume the dimension of input prompt embeddings is $[B, N, L, C]$, where $B$, $N$, $L$, and $C$ denote batch size, length of prompt batch (*i.e.*, number of prompts), sequence length, and hidden dimension, respectively. The cascaded convolutional blocks transform prompt embeddings to match the dimensions of tokenized weights. Here, we refer the output of the last block as $[B, N_w, L_w, C_w]$.

**Learning objective.** The learning objective is simple: we calculate the mean squared error (MSE) loss between the output of the last block of the parameter generator and the corresponding tokenized checkpoints. Similar to RPG [58], we tokenize each layer's parameters into non-overlapping segments and apply padding, ensuring checkpoints have a consistent shape of $[B, N_w, L_w, C_w]$. Formally, we write the MSE loss below,

$$\text{prompt embeddings} \xrightarrow{\text{parameter generator}} L_{MSE} \xleftarrow{\text{tokenization [58, 4]}} \text{corresponding checkpoints.} \quad (4)$$

**Inference.** We expect the parameter generator to develop effective "drag-and-drop" ability, particularly for novel datasets or tasks not encountered during training. Therefore, our evaluation primarily focuses on performance across novel datasets. The inference process consists of four steps: 1) sampling prompts from novel datasets, 2) extracting embeddings from these prompts, 3) feeding the embeddings into the parameter generator, and 4) evaluating the generated parameters on the novel datasets. To comprehensively demonstrate the "drag-and-drop" ability of our approach, we examine performance in both in-domain scenarios (*e.g.*, common sense-to-common sense) and cross-domain scenarios (*e.g.*, common sense-to-science).

## 3 Experiments

### 3.1 Implementation Details

We choose Qwen2.5 [63] series as foundation model and conduct experiments on common sense reasoning, coding, math, and multimodal tasks. The details of involved model sizes and datasets for each task are listed in the table below. The default text encoder is Sentence-BERT [47], and length of prompt batch is set to 128, 64, 64 and 32 for common sense reasoning, math, coding, and multimodal task, respectively. For other hyper-parameter settings, please refer to Appendix A.1.

| task | #model size (B) | datasets |
|---|---|---|
| common sense | 0.5 | ARC-e [11], ARC-c [11], BoolQ [10], OBQA [41], HelaSwag [65], PIQA [6], WinoGrande [50] |
| coding | 1.5, 7 | Evol-Instruct-68K-V1 [40], Glaive-Assistant-V2 [18], Python-Codes-25K [16], Code-74k-ShareGPT [2], Rosetta-Code [13], LLaMA-Python-Codes-30K [15], CodeAlpaca-20K [8] |
| math | 1.5 | Competition-Math[24], Math-QA[3], Math-IIO-68K-Mini [45] Math-Plus [56], Mu-Math [64], ToT-Math-V1 [42] |
| multimodal | 3 | MathV360K [55] |

### 3.2 Common Sense Reasoning

**Evaluating setting.** We employ LoRA [25] to fine-tune Qwen2.5-0.5B on seven common sense reasoning datasets and save the checkpoints as training data. In every column of Table 1, we use the specified dataset as test set (*i.e.*, not used in training) and train DnD on other datasets' LoRAs.

| method \ test set | ARC-e | OBQA | ARC-c | PIQA | HellaSwag | BoolQ | WinoGrande |
|---|---|---|---|---|---|---|---|
| training LoRAs | 37.5 | 30.2 | 39.5 | 40.5 | 22.4 | 13.5 | 38.8 |
| **DnD** | **68.6** | **40.8** | **51.6** | **87.9** | **25.9** | **44.9** | **50.0** |
| average accuracy improvement: 21.0 on training LoRAs | | | | | | | |

Table 1: **Generalization on novel (test) datasets**. Our approach significantly outperforms LoRAs used in training in terms of accuracy (%) across *all* unseen datasets. **Bold entries** are the best results.

**Analysis.** We report the average accuracy of training LoRAs and our generated ones in Table 1. Several observations can be made from these results: i) Our method consistently outperforms LoRAs used for training on unseen datasets, indicating it manages to drag-and-drop LLM parameters to task-specific distribution specified via condition. ii) This drag-and-drop ability holds across different datasets, showing strong robustness towards various data inspirations.

**Cross-domain Drag-and-Drop.** To further explore DnD's zero-shot ability, we not only use in-domain novel datasets in inference, but also test it on cross-domain task. In this part, we use the checkpoint trained on common sense reasoning tasks and feed it with prompts from science-dataset [34]. The generated parameters are compared with training LoRAs on the science dataset. From Table 2, we can observe that DnD surpasses its training LoRAs' average accuracy,

| testset | training LoRAs | **DnD** | improvement (↑) |
|---|---|---|---|
| science | 35.6 | **45.3** | 9.7 |

Table 2: Our approach also succeeds on cross-domain scenario (*i.e.*, both novel data and task).

indicating our method manages to drag-and-drop LLMs for cross-domain tasks (*i.e.*, from common sense reasoning to science). Note training LoRAs are trained on ARC-e, OBQA, PIQA, HellaSwag, WinoGrande, BoolQ, and they've never seen a science dataset.

| method \ task | Coding (HumanEval) | | | Math | | Multimodal | |
|---|---|---|---|---|---|---|---|
| | pass@1 | pass@5 | pass@10 | gsm8K | MATH | Math-Vision | Math-Vista |
| training LoRAs | 17.6 | 28.6 | 33.2 | 42.9 | 14.8 | 23.0 | 61.5 |
| **DnD** | **32.7** | **55.3** | **64.1** | **66.3** | **23.9** | **24.3** | **62.3** |
| improvement (↑) | 15.1 | 26.7 | 30.9 | 23.4 | 9.1 | 1.3 | 0.8 |

Table 3: DnD can generate parameters for more complex tasks like math, code and multimodal question answering. Our method continues to show strong zero-shot ability on these tasks.

## 3.3 Generalization to coding, math, and multimodal tasks

To further validate our method's applicability in more complex scenarios, we also employ DnD for coding, math, and multimodal tasks. The empirical results and findings are as belows.

**Coding.** Similar to common sense reasoning task, we use LoRA to fine-tune Qwen2.5-1.5B on seven coding datasets and save the checkpoints as training data. Evaluation is carried out on HumanEval [9] benchmark using pass@k [33] score (k = 1, 5, 10). *Note that neither LoRA fine-tuned models nor DnD has seen any samples of the benchmark in their training*. Therefore, we directly test training LoRAs and our synthesized ones on HumanEval. From Table 3, we can draw some findings: i) Our method yields promising results, with improvement over average **pass@1 = 15.1, pass@5 = 26.7, pass@10 = 30.9**. ii) Despite training LoRAs perform poorly on the test set, DnD still obtains good performance. *This means instead of memorizing parameters seen in training, it learns to fit novel datasets given condition as inspirations*.

**Math.** We fine-tune Qwen2.5-1.5B on six math datasets and save the checkpoints. We adopt gsm8K [12] and MATH [24] as our benchmarks and accuracy as evaluation metrics. Results listed in Table 3 show commonalities with those regarding common sense reasoning and coding tasks, underscoring the superiority of our method across a wide range of scenarios.

**Multimodal.** The above results confirm our method's effectiveness in text modality. In the following, we explore its greater potential by pacing towards other modalities. We fine-tune Qwen2.5-VL-3B [5] on MathV360K [55], save the checkpoints, and evaluate using Math-Vision [60] and Math-Vista [39]. Results in Table 3 show that DnD perform well in multimodal tasks, revealing that *our method can be adapted to modalities other than texts* and has promising application potential.

**Takeaway:** Based on the above results and comparisons, DnD is a *high-performing zero-shot learner* with strong robustness and wide applicability, as reflected by the *significant improvements* compared to its training data and its promising performance *across various scenarios*. In the following, we continue to explore more interesting features of our proposed approach.

## 3.4 Ablation Studies

This section mainly aims at exploring a series of interesting features about our approach. For those exploring various settings in our experiment, we report them in Appendix B.3 for thoroughness. If not stated, we use ARC-c as the test set and other common sense reasoning datasets for training.

**What types of data will help Drag-and-Drop LLMs better?** As introduced in Section 2.3, we use prompts as conditions to inspire DnD. Can this drag-and-drop ability holds when condition types changes (*e.g.*, answers)? We carry out ablation studies by changing condition types as prompt, prompt + answer and their mixture (prompt : answer=4 : 1) and report the results in Table 4a.

It can be observed that the prompt + answer group surprisingly lead to poor performance. We conclude that it is because answers in common sense reasoning datasets lack diversity (*i.e.*, A/B/C/D) and combining them with prompts may detriment dataset-specific representations. This shall hinder generator to distinguish different datasets and generate specific parameters. Consequently, we advise not to use answer alone as inspirations. However, conclusions may be different for some tasks where answers are more complicated and diverse and we show the results on these tasks in Appendix B.3. **How does the choice of condition extractor affect DnD's performance?** Our default condition extractor is Sentence-BERT [47], yet it is interesting to explore other models' potentials. To ensure

| condition type | accuracy |
|---|---|
| prompt | **51.6** |
| prompt + answer | 27.0 |
| mix | 49.7 |
| training LoRAs | 39.5 |

(a) **Condition types.** Pure prompts used as inspirations yield the best results compared to other formats.

| condition extractor | accuracy |
|---|---|
| Glove | 50.8 |
| Sentence-BERT | **51.6** |
| T5-base | 50.2 |
| Qwen2.5-7B | fail |

(b) **Extractor structure.** Several encoder-based extractors perform better than decoder-only ones.

| dataset arrangement | improves. |
|---|---|
| $6 \in$ train, $1 \in$ test | **12.1** |
| $4 \in$ train, $3 \in$ test | 11.4 |
| $3 \in$ train, $4 \in$ test | 0.8 |
| $2 \in$ train, $5 \in$ test | -1.4 |

(c) **Train-test set arrangements.** More diverse training data introduces higher improvements.

| length | DnD | improves. |
|---|---|---|
| 8 | 33.2 | -6.3 |
| 32 | 47.3 | 7.8 |
| 128 | **51.6** | 12.1 |

(d) **Length of prompt batch.** As prompt batch grows longer, DnD's performance improves accordingly.

| method | w/o cvg. | cvg. |
|---|---|---|
| training LoRAs | 27.3 | 39.5 |
| DnD | 46.3 | **51.6** |
| improvement (↑) | 19.0 | 12.1 |

(e) **Checkpoint quality.** 'cvg.' denotes 'converged'. DnD maintains strong even with weak LoRAs.

| method | full | 10% |
|---|---|---|
| training LoRAs | 39.5 | 29.8 |
| DnD | **51.6** | 49.1 |
| improvement (↑) | 12.1 | 19.3 |

(f) **Dataset size.** DnD maintains strong even with only 10% data (51.6%→ 49.1%).

Table 4: Ablation studies about condition types, condition extractor's type, train-test set arrangement, length of prompt batch, checkpoint quality, and dataset size. This series of explorations validate several designs of DnD.

thorough comparisons, we include classical word2vector method Glove [44], the default encoder-only Sentence-BERT [47], encoder-decoder model T5 [46] and decoder-only Qwen2.5-7B [63].

Results in Table 4b reveal several insights: i) Even traditional methods such as Glove can help DnD to obtain promising result, indicating our method can fit plenty of text encoders. ii) Qwen2.5-7B performs not as good as expect, which has two possible causes: First, its heaviness limits the number of conditions paired with parameters per iteration, leading to poor awareness of novel datasets. Similar conclusions can be drawn from our experiments in Appendix B.3. Second, Qwen2.5-7B's decoder-only architecture may constraint conditions' diversity, since it encodes prompts to answers.

**What property of training data equips our method with drag-and-drop ability?** By default, we train on several datasets and test on 1 novel dataset. In this part, we explore DnD's robustness by shrinking the number of train sets and test on more datasets. Train-test set arrangements are: 6-1, 4-3, 3-4 and 2-5. Generated parameters are compared with training LoRAs' average accuracy on the unseen datasets and their average improvements in accuracy are reported in Table 4c.

It can be observed that: i) Generally, more training datasets lead to better performance improvement. This is expected since more data ensures better coverage of condition-parameter correlations and lead to better robustness for novel data. ii) DnD fails to drag-and-drop LLMs to novel datasets when training samples are few. As datasets used for training lessen, the average improvement of DnD drops accordingly. It hardly improves over training LoRAs for the 2-5 case. We can conclude that basic amount of training samples are needed for DnD to learn condition-parameter correlations.

**How does DnD's performance compared with foundation LLMs?** Given massive pretraining LLMs often take, fine-tuning on a small downstream dataset may detriment their zero-shot performance on novel test sets. Aware of this phenomenon, we compare DnD generated weights' performance with foundation LLMs across all tasks involved in our experiment. Specifically, for foundation LLMs, we adopt Qwen2.5-0.5B

| testset \ method | foundation LLM | **DnD** | improves. (↑) |
|---|---|---|---|
| ARC-c | 38.3 | **51.6** | 13.3 |
| HumanEval | 32.3 | **64.1** | 31.8 |
| gsm8K | 64.4 | **66.3** | 1.9 |
| Math-Vision | 22.7 | **24.3** | 1.6 |

Table 5: DnD surpasses foundation LLMs across various tasks, showing the "drag-and-drop" effect.

for common sense reasoning, 1.5B for math and coding, and Qwen2.5-VL-3B for multimodal task. Results in Table 5 again show our approach's superiority: DnD outperforms foundation LLMs across all tasks. Its "drag-and-drop" ability can generate task-specific parameters, with performance better than foundation LLMs that go through abundant pretraining.

**How does the length of prompt batch impact DnD?** We change the length of prompt batch, turning it to 8, 32, 128 and explore their influence on DnD. From Table 4d, when prompt batch is too small (e.g., 8), DnD's performance is not good. but can surpass training LoRAs with larger prompt batch (e.g. $\geq 32$). As prompt batch grows longer, DnD's performance improves accordingly.

**How does the quality of LoRAs impact DnD?** In addition to collecting well-trained checkpoints, we collect checkpoints at the beginning of training trajectory, i.e., checkpoints not converged on the training sets. As shown in Table 4e, DnD maintains good performance even with worse-performing LoRAs. Its performance drops with training LoRAs that are insufficiently tuned, but still surpasses converged training LoRAs on zero-shot test set. Therefore, DnD works well with worse training LoRAs, which reveals that our method has strong data adapting ability.

**How does dataset size impact LoRA training?** In our paper, we use the entire dataset to train and collect LoRA checkpoints. To ablate the influence of dataset size, we use 10% of each dataset to train LoRAs, then adopt these LoRAs for DnD's training, and evaluate their performance. Based on the results in Table 4f, even trained with only 10% data, DnD still maintains good performance ($51.6\% \rightarrow 49.1\%$). As expected, training LoRAs' performance drops largely ($39.5\% \rightarrow 29.8\%$) with only 10% training data. DnD' performance drop is much smaller than training LoRAs on novel datasets (i.e., ARC-c). Therefore, DnD is less sensitive to dataset size than traditional LoRA tuning, demonstrating its strong dataset-adapting ability.

### 3.5 Open Explorations and Analysis

**Condition-Parameter Pairing.** In this part, we explore other pairing strategies' influence on performance than that introduced in Section 2.3. We test 2 condition pairing strategies:

- **Strategy 1:** We fix the total number of prompts to be 128, 256, 512, 1024, 2048, 5000 and use all those prompts to pair with parameters every iteration ($x \leftarrow x$).

- **Strategy 2:** We fix the length of prompt batch to be 128 in every iteration and randomly picks these prompts from 128, 256, 512, 1024, 2048, 5000 candidate prompts ($128 \leftarrow x$).

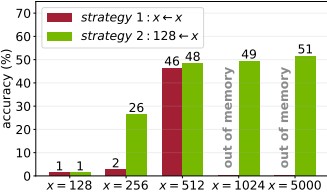
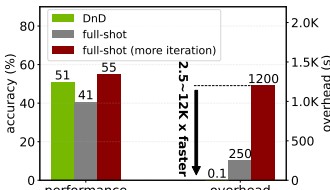
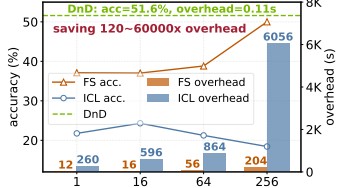

(a) Random selection and pairing is better than using the same conditions for all parameters.

(b) DnD can reach comparable or even better performance than full-shot while being 2.5~12K × faster.

(c) DnD outperforms popular few-shot tuning and ICL before 256 shots while avoiding using answers.

Figure 4: Explorations about DnD's condition-parameter pairing strategy, compare DnD with state-of-the-art methods, and its superiority over few-shot tuning and in-context learning.

Based on the results in Figure 4a, we can draw several conclusions: i) With limited number of conditions, DnD fails to generalize over novel data, since it can hardly learn comprehensive knowledge about condition-parameter mapping's landscape. ii) As number of condition increases, **Strategy 2**'s performance skyrockets since DnD is exposed to sufficient condition-parameter pairs. This indicates **Strategy 2** may help DnD to converge efficiently. iii) **Strategy 1** needs more conditions to reach comparable performance as **Strategy 2**. With large number of conditions, **Strategy 1** suffers from out of memory issues. The same-condition-per-parameter strategy may hinder DnD to associate conditions with specific datasets. Conclusively, **Strategy 2** is superior than **Strategy 1** both in model generality, convergence speed, and memory consumption. These conclusions are consistent with findings in Table 4c: *Diversity of training data equips our method with drag-and-drop ability*.

**DnD vs full-shot tuning.** In this part, we compare DnD with full-shot tuning in accuracy and overhead. Specifically, we test ARC-c tuned LoRA's ($\approx$75 iterations for ARC-c, detailed in Appendix A.4) full-shot performance using ARC-c's test set. Since LoRA's performance might improve with more iterations, we fine-tune LoRA on ARC-c for 300 iterations and test its performance as well. These results are compared with **zero-shot** DnD in Figure 4b: i) using mild tuned checkpoints for training, DnD already yields better results than full-shot tuning. This showcases DnD's impressive **zero-shot** ability even surpasses **full-shot tuning**. ii) DnD is incredibly efficient, with better performance than full-shot tuning while being **2500** × faster. Moreover, as training continues, though full-shot prevails, our method shows minimal performance gap with it while being **12,000** × more efficient.

**Efficiency analysis of DnD.** In additional to full-shot tuning, in-context learning (ICL) and few-shot tuning (FS) are also popular methods in LLM fine-tuning. In Figure 4c, we conduct experiments

investigating the performance-efficiency trade-off of them. Several observations can be drawn: i) Both ICL and FS' results are poor when shots are few, but their overhead rises as shots increase. ii) DnD can reach better performance than FS and ICL before 256 shots with negligible overhead. iii) *It is noteworthy that both few-shot and ICL use answers for instructing the LLM* to obtain better performance. On the contrary, DnD relies on only 128 unlabeled prompts. Based on the above results, we anticipate *DnD is a powerful and efficient zero-shot learner*.

**Scalability of DnD.** In this part, we explore DnD's scalability. Since common sense reasoning is simple and 0.5B model suffices, we focus on math and coding tasks while increasing foundation model size from 1.5B to 7B. We use original math and coding datasets in Section 3.3, evaluate using gsm8K for math and more diffi-cult benchmark LiveCodeBench [26] for coding.

| testset \ method | training LoRAs | **DnD** | improves (↑) |
|---|---|---|---|
| LiveCodeBench | 13.0 | **33.3** | 20.3 |
| gsm8K | 65.9 | **83.1** | 17.2 |

Table 6: DnD scales well with larger 7B foundation models, and maintains strong performance in more complex benchmark LiveCodeBench.

We report accuracy on math task and pass@1 for coding in Table 6. It can be observed that: i) DnD consistently surpasses training LoRAs in both tasks under 7B model setting, underscoring its eminent scalability for larger foundation LLMs. ii) With more difficult coding benchmark, DnD maintains superior performance with training LoRAs, *i.e.*, improvement over average **pass@1 = 20.3**. It reveals DnD's capacity for generalizing to more complex benchmarks, showing promising application potential and robustness.

**Comparisons with previous methods.** We compare our method with most recent parame-ter generation method RPG [58]. We explore both methods' perfor-mance in two scenarios: Close set generation: generate param-eters seen in training. Open set generation: generate parameters for unseen datasets. Figure 5 shows both methods work well on close set generation, but RPG fails on

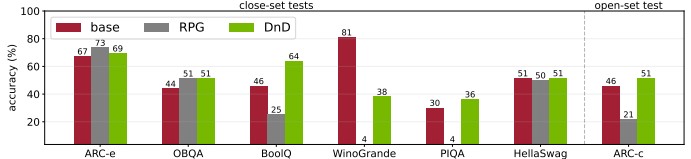

Figure 5: DnD and RPG perform well in most close-set tests. However, RPG can hardly generate parameters for novel dataset while DnD still presents strong zero-shot ability on open-set test.

open set generation, showing that our designs (prompts as conditions, condition-parameter pairing) have certain robustness and generality towards novel datasets.

**Visualization for the effect of drag-and-drop.** In this part, we visualize original and generated parameters in Figure 6. It can be seen that original parameters exhibit diverse patterns in the weight space, forming different clusters. Moreover, even param-eters close to those tuned on the target dataset (*i.e.*, ARC-c) can have large performance gap (*i.e.*, 19.1% compared with 40.7%). After training on these models that have distinct features, DnD

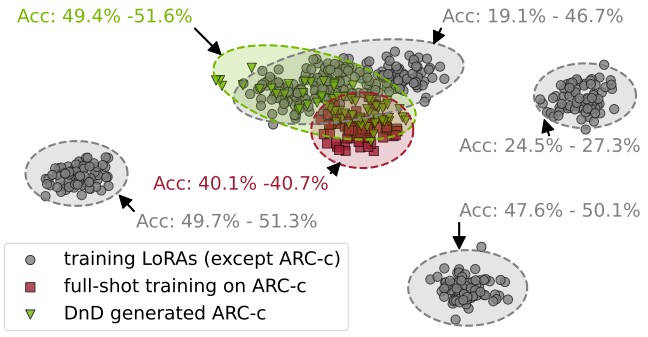

Figure 6: DnD generates parameters with close distribution to original ones in the weight space and promising performance.

can generate parameters for the target dataset in a zero-shot manner. The generated parameters are close to the original ones in the weight space and with even superior performance than full-shot tuning (*i.e.*, 51.6% compared with 40.7%). This brings the "drag-and-drop" effect to life.

**Exploration of the domain gap influence.** Without loss of generality, we test cross-domain with a large domain gap, i.e., coding-to-math. We use datasets and checkpoints in math and coding tasks to train DnD jointly. Based on the results in Table 7, DnD improves largely over its training LoRAs on gsm8K, showing it has the potential of cross complex domains.

| method | accuracy (%) |
|---|---|
| training LoRAs | 46.3 |
| DnD with joint training | 65.3 |
| improvement (↑) | 19.0 |

Table 7: Performance on gsm8K: train-ing LoRAs vs. DnD with joint training.

## 4 Related Works

**Parameter-Efficient-Fine-Tuning (PEFT).** LLMs' size rapidly scales up in the 2020s, making full parameter fine-tuning infeasible. To address this, Low-Rank Adaptation (LoRA) [25] has been proposed, leveraging LLMs' inherent sparsity and substantially reduces fine-tuning costs by optimizing two low rank matrices instead of the original weights. Multiple variants of LoRA emerge afterwards, such as DoRA [37], LoRA+ [23], VeRA [31], and DyLoRA [57]. However, all of them have one potential flaw: they requires tuning model parameters for every novel dataset, therefore lacks generality. This can still induce extra costs as model size enlarges and training data increases.

**Parameter generation.** Parameter generation trains on model checkpoints and aims at generate high-performing parameters, both seen and unseen in training. Previous works [7, 20, 30, 17] focuses on learning distributions over the parameters, yet struggle to reconstruct original models' performance. With the development of diffusion models, Hyper-Representations [51, 52, 54] and p-diff [59], use the latent diffusion architecture to generate high-performing parameters. Armed with Mamba [21] and appropriate tokenization strategy, RPG [58] can generate 200M of parameters in minutes. Regarding conditional generation, COND P-DIFF [27], Tina [35] and ORAL [28] explores text-controlled parameter generation method. RPG even generate parameters for novel datasets on binary embedding classification task for CIFAR-10 [32]. However, these methods can hardly keep promising zero-shot ability on more complex tasks, hindering parameter generation's greater potential. On the other hand, our method use prompts in novel datasets as conditions, captures parameters' relations with datasets better, and is able to generate competent parameters for novel datasets.

## 5 Discussion and Conclusion

PEFT offers an efficient solution to reduce costs when customizing LLMs/VLLMs for downstream tasks, yet it is still expensive when models are large and tasks are diverse. In our study, we train a parameter generator to map prompt-weight pairs, which can produce customized weights for novel tasks by processing unlabeled task prompts. Notably, our approach can transform *unlabeled task prompts* directly to LoRA weights update *in seconds*. This prompt-to-weight paradigm, which can generate task-specific weights *without further tuning*, sheds lights on a promising new direction for efficient LLM and VLLM customization.

Previous research [51, 59, 58] and our approach demonstrate that neural network weights can be effectively synthesized. It appears that network parameters can be viewed as a new form of data modality. To excavate this emerging field's potential, several challenges remain to be tackled. First, scaling parameter generation to larger models (7B-70B parameters) requires novel architectural and algorithmic advances. Second, leveraging existing pre-trained checkpoints from the Internet could enhance the practicality of parameter generators. Third, generating structurally diverse models adaptable to various hardware configurations would improve deployment flexibility.

## Acknowledgments and Disclosure of Funding

We sincerely appreciate Yuxiang Li, Jiaxin Wu, Zhiheng Chen, Lei Feng, Jingle Fu, Hesen Yang, Bohan Zhuang, Ziheng Qin, Zangwei Zheng, Zihan Qiu, Zexi Li, Gongfan Fang, Xinyin Ma, and Qinglin Lu for valuable discussions and feedbacks during this work.

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

We organize our appendix as follows.

**Hyper-Parameter Settings**

**Additional Experiment Results**

# A  Hyper-Parameter Settings

## A.1  Training Recipe

In this section, we provide details of our training recipe and various hyper-parameter settings. We incorporate multiple tasks in language models, each involves different foundation model sizes, different generator architecture, and training schedules. We report settings for every task in Table 8.

| training setting | common sense | coding | math | multimodal |
|---|---|---|---|---|
| batch size | 128 (0.5B) | 128 (1.5B), 48 (7B) | 128 (1.5B), 48 (7B) | 64 (3B) |
| optimizer | AdamW | AdamW | AdamW | AdamW |
| learning rate | 3e-5 | 3e-5 | 3e-5 | 3e-5 |
| length of prompt batch | 128 | 16 | 32 | 16 |
| training step | 5,000 | 5,000 | 5,000 | 5,000 |
| weight decay | 0.1 | 0.1 | 0.1 | 0.1 |
| max grad norm | 1.0 | 1.0 | 1.0 | 1.0 |
| noise aug. amplitude | 1e-4 | 1e-4 | 1e-4 | 1e-4 |

Table 8: Training recipe for different tasks in Section 3.2 and Section 3.3.

**Length of prompt batch.** It has been introduced in Section 2.3 that every parameter is grouped with certain amount of texts each iteration. Due to the length of prompt in different datasets varies and induces variable training costs, the length of prompt batch also varies.

**Revision statement.** During preparation of camera ready version, we fix some minors of the code and list them here: 1) the tokenization process, 2) the warm up scheduler. We updated the code to our GitHub repository.

## A.2  Description of Datasets

In this section, we introduce the datasets used in the paper, including those for common sense reasoning, math, coding, and multimodal tasks.

**Common sense reasoning. ARC** dataset [11] contains grade-school level, multiple-choice science questions and is splited into easy and challenge sets. **OBQA** [41] aims to promote research in advanced question-answering with salient facts summarized as an open book. **PIQA** [6] focuses on everyday situations with a preference for atypical solutions. **HellaSwag** [65] instructs models to select from choices that best finish the sentence among ground truth and an adversarial set of machine-generated wrong answers. **WinoGrande** [50] features a fill-in-a-blank task with binary options for commonsense reasoning questions. **BoolQ** [10] is a question answering dataset for yes/no questions containing various factual problems.

**Coding. Evol-Instruct** [40] contains evolutionary prompts tailored for code-related tasks and incorporates code debugging and time-space complexity constraints. **Glaive-Assistant** [18] is a dataset of code problems and solutions generated using Glaive's synthetic data generation platform. **Python-Codes** [16] is a cleaned Python dataset covering instructional tasks. **Code-ShareGPT** [2]

consists of conversations along with detailed Python code explanations. It is generated using GPT-3.5, GPT-4 etc. **Rosetta-Code** [13] presents solutions to the same task in as many different languages as possible, to aid a person with a grounding in one approach to a problem in learning another. **LLaMA-Python-Codes** [15] primarily focuses on instructional tasks in Python, tokenized specifically for the Llama architecture. It is a blend of GPT-4 generated content, custom codes, behavioral approaches and tasks. **Code-Alpaca** [8] dataset is generated by the techniques in [61], with some modifications. **Math. Competition-Math** [24] consists of problems from math competitions, including the AMC 10, AMC 12, AIME, and more. **Math-IIO-68K-Mini** [45] mathematical questions and with corresponding step-by-step solutions. **MathQA** [3] is a large-scale dataset of math word problems that are densely annotated with operation programs. **Math-Plus** [56] is a augmented dataset with GPT-4. **Mu-Math** [64] is a meta-evaluation dataset derived from the U-MATH [64] benchmark, intended to assess the ability of LLMs to judge free-form mathematical solutions. **ToT-Math-V1** [42] prioritizes reasoning and explanatory problem-solving over provided answers.

**Multimodal.** MathV360K [55] consists 40K images from 24 datasets and 360K question-answer pairs. It is used to enhance the multimodal mathematical reasoning capabilities of MLLMs.

### A.3  Detailed Structure of Hyper-Convolutional Decoder

**Details of condition extractor.**  As introduced in Section 3, we use Sentence-BERT [47] as our condition extractor in default (all-MiniLM-L6-v2 specifically). Since BERT's supported sequence length is only 512, for longer sequences, we need to preprocess the input sequences. Specifically, we first pad the sequence to the length that can be divided by 512, slice it into multiple sub-strings, and encode the sub-strings respectively. The input length for different tasks is in the table below.

|        | common sense | coding | math | multimodal |
|--------|--------------|--------|------|------------|
| length | 384          | 26624  | 4608 | 1536       |

**Details of hyper-convolutional decoder.**

In this part, we delve into hyper-convolutional decoder's inner architecture introduced in Section 2.5. The decoder consists of multiple cascading hyper-convolutional blocks, each containing 5 2D convolution modules. Specifically, we divide convolutions into three categories: i) **width convolution** that operates on $(C, L)$ dimension, ii) **height convolution** that operates on $(L, N)$ dimension) iii) **layer-wise convolution** that on $(N, L)$ dimension) , with notations $\text{Conv}_W$, $\text{Conv}_H$, and $\text{Conv}_L$. In the above convolution modules, the input tensor is transposed to the shape that specified dimensions act as feature maps, the remaining dimension act as channel dimension like conventional convolution. Each hyper-convolutional block consists of two $\text{Conv}_W$, two $\text{Conv}_H$ and one $\text{Conv}_L$. Given this, the forward operation of a hyper-convolutional block can be formulated as:

$$
\begin{aligned}
c_W^l &= \text{Conv}_H^1(\text{Conv}_W^1(c^{l-1})); \\
c_H^l &= \text{Conv}_W^2(\text{Conv}_H^2(c^{l-1})); \\
c^l &= \text{Conv}_L(\frac{c_W^l + c_H^l + b}{3}),
\end{aligned}
\tag{5}
$$

where $c^l$ is hidden state output by the $l$ th layer, $c^0$ is prompt embedding encoded by the condition extractor, and $b$ is learnable bias. Through this process, it transforms the input with shape of $[B, N, L, C]$ to $[B, N', L', C']$.

**Model architecture used in different tasks.**  In this part, we show the architecture of hyper-convolutional decoders. We use the three element tuple $(N, L, C)$ to represent decoder structure since it reflects input conditions' changes in the network. Also, $B$ dimension is omitted since it doesn't affect the model architecture. Note that for math and coding tasks, we first project the $L$ dimension to 1000 to reduce overwhelming memory costs induced by giant convolution kernels.

### A.4  Details of Trained Checkpoints Collection

In this section, we discuss how we collect checkpoints in detail. It is noteworthy that all checkpoint collection process go through two phases: i) Pretraining on the target dataset for specified steps. ii) Fine-tuning on the target dataset for certain additional steps, while saving a checkpoint at each step.

Therefore, the essence of checkpoint collection is the pretrain and fine-tune phase's learning rate, training steps, number of samples, and batch size. Note that except for learning rate and training

| task | foundation model | channel |
|---|---|---|
| common sense | 0.5B | (128, 384, 384)→(128, 200, 300)→(128, 100, 256)
(256, 50, 200)→(512, 50, 200)→(1024, 25, 200)
(1024, 10, 200)→(2048, 10, 200)→(4296, 8, 128) |
| coding | 1.5B | (32, 1000, 384)→(64, 500, 300)→(256, 500, 300)→(512, 125, 300)
(1024, 64, 256)→(2048, 32, 256)→(4508, 16, 256) |
| | 7B | (16, 1000, 384)→(64, 500, 384)→(256, 125, 400)→(512, 64, 400)
(1024, 64, 400)→(2048, 32, 400), (4928, 16, 512) |
| math | 1.5B | (16, 1000, 384)→(64, 500, 300)→(256, 125, 300)→(512, 64, 300)
(1024, 64, 256)→(2048, 32, 256)→(4508, 16, 256), |
| | 7B | (16, 1000, 384)→(64, 500, 384)→(256, 125, 400)→(512, 64, 400)
(1024, 64, 400)→(2048, 32, 400), (4928, 16, 512) |
| multimodal | 3B | (16, 1536, 384)→(64, 500, 300)→(256, 125, 300)
(1024, 64, 300)→(2048, 16, 256)→(7308, 16, 256) |

Table 9: Model architectures for different tasks.

steps, all other hyper-parameters are kept the same. Detailed settings are in Table 10. For those datasets contain less samples than the number specified below, we use the entire dataset for training.

| task \ setting | pretrain | | | | finetune | |
|---|---|---|---|---|---|---|
| | lr. | training step | batch size | #num. samples | lr. | training step |
| common sense | 1e-4 | 75 | 32 | 5000 | 1e-5 | 50 |
| coding | 1e-4 | 4000 | 64 | 10000 | 1e-6 | 100 |
| math | 1e-4 | 4000 | 64 | 10000 | 1e-6 | 100 |
| multimodal | 1e-4 | 8000 | 64 | 100000 | 1e-6 | 200 |

Table 10: Details for checkpoint collection.

The number of checkpoints used for each dataset during model training impacts the learning dynamics of mapping data to weights. Empirical experiments revealed that (1) insufficient checkpoints may hinder the model's ability to capture nuanced parameter interactions, while (2) excessive checkpoints increase computational overhead without proportional gains. To balance efficiency and performance, we configured checkpoint counts as follows.

| task | number of checkpoints |
|---|---|
| common sense reasoning | 50 |
| coding/math 1.5B, math 7B | 80 |
| coding 7B | 100 |
| multimodal 3B | 200 |

Table 11: Number of checkpoints per dataset, selected to balance effectiveness and training cost.

# B  Additional Experiment Results

## B.1  More Results of Common Sense Reasoning, Math, and Coding

In this section, we delve into details about each group of training LoRAs' performance on test sets, report their performance and discuss further findings.

**Common sense reasoning.** In this part, we show training LoRAs' performance on each test set in Table 12. Each row specifies the dataset LoRA is trained on, and each column denotes the dataset used for testing. Consequently, the diagonal elements are full-shot results of training LoRAs, which are marked in red . We also incorporate their average accuracy (exclude diagonal elements, marked in gray ), Qwen2.5-0.5B's performance, and DnD's zero-shot results (marked in green ).

| train set \ test set | ARC-e | OBQA | ARC-c | PIQA | HellaSwag | BoolQ | WinoGrande |
|---|---|---|---|---|---|---|---|
| training LoRA of ARC-e | 59.4 | 46.2 | 46.7 | 80.4 | 30.6 | 44.6 | 52.2 |
| of OBQA | 64.3 | 38.4 | 53.4 | 56.1 | 29.0 | 1.3 | - |
| of ARC-c | 57.2 | 38.2 | 40.7 | 65.9 | 46.7 | 23.6 | - |
| of PIQA | 27.9 | 27.0 | 24.7 | 66.2 | 24.4 | 9.2 | 50.3 |
| of HellaSwag | 57.2 | 43.2 | 41.0 | 40.4 | 23.4 | 0.5 | 52.8 |
| of BoolQ | fail | fail | 52.0 | fail | fail | 22.11 | fail |
| of WinoGrande | 18.6 | 26.8 | 19.1 | 0.1 | 3.8 | 1.6 | 50.5 |
| Qwen2.5-0.5B | 54.8 | 16.6 | 38.3 | 16.6 | 26.5 | 37.0 | 50.2 |
| average of training LoRAs | 37.5 | 30.2 | 39.5 | 40.5 | 22.4 | 13.5 | 38.8 |
| DnD | 68.6 | 40.8 | 51.6 | 87.9 | 25.9 | 44.9 | 50.0 |

Table 12: More results for common sense reasoning task. Red marks full-shot results, gray shows the average of training LoRAs, and green marks the results of DnD. DnD not only consistently surpasses average of training LoRAs, but also outperform their full-shot performance in most cases.

It can be observed that: i) original LoRAs generally perform poorly on novel datasets, *e.g.*, LoRAs tuned on BoolQ fail (accuracy=0.0) on half of zero-shot datasets, this may because training on one specific dataset will fit the parameters for this dataset (binary True/False in BoolQ's case) and limit their generality (multiple choices question of other datasets). ii) DnD even outperforms training LoRAs' full-shot performances in most cases, showing it has incredible zero-shot ability with great efficiency. These findings further illustrates that: *fitting parameters for certain data may lack generality, learning the correlations between data and parameters may be better.*

**Coding.** In this part, we elaborate on training LoRAs' performance on coding datasets, along with their average (marked in gray ), Qwen2.5-1.5B/7B, and DnD's performance (marked in green ).

| test set \ train set | Share GPT | Evol Instruct | Glaive Assistant | Python | Rosetta | LLaMA Python | Alpaca | Qwen2.5 1.5B | train set avg. | DnD |
|---|---|---|---|---|---|---|---|---|---|---|
| pass@1 | 28.8 | 40.0 | 14.0 | 9.8 | 17.6 | 13.7 | 23.2 | 14.7 | 17.6 | 32.7 |
| pass@5 | 46.2 | 53.4 | 27.8 | 19.1 | 28.6 | 25.5 | 31.0 | 26.5 | 28.6 | 55.3 |
| pass@10 | 52.9 | 56.4 | 35.0 | 23.8 | 33.2 | 29.9 | 34.1 | 32.3 | 33.2 | 64.1 |
| testset: **HumanEval**, average improvement: pass@1 = **15.1**, pass@5 = **26.7**, pass@10 = **30.9** | | | | | | | | | | |

(a) pass@k (k = 1, 5, 10) scores of foundation LLM, original and generated LoRA for Qwen-1.5B on HumanEval. DnD shows improves largely over its training data and base Qwen, validating its effectiveness.

| test set \ train set | Share GPT | Evol Instruct | Glaive Assistant | Python | Rosetta | LLaMA Python | Alpaca | Qwen2.5 7B | train set avg. | DnD |
|---|---|---|---|---|---|---|---|---|---|---|
| pass@1 | 22.4 | 23.2 | 24.6 | fail | fail | fail | fail | 34.1 | 13.0 | 33.4 |
| pass@5 | 28.4 | 28.2 | 32.7 | fail | fail | fail | fail | 41.6 | 16.4 | 42.1 |
| pass@10 | 30.8 | 30.3 | 35.8 | fail | fail | fail | fail | 43.8 | 17.6 | 46.0 |
| testset: **LiveCodeBench**, average improvement: pass@1 = **20.3**, pass@5 = **25.7**, pass@10 = **28.4** | | | | | | | | | | |

(b) DnD constantly present promising results at more complex benchmarks (*i.e.*, LiveCodeBench), even when over half of its training LoRAs fail (pass@k = 0.0) on this dataset.

Table 13: DnD surpasses the average pass@k score of training data on zero-shot coding benchmarks.

Empirical results in Table 13 indicates that: i) our approach is able to generalize to complex real-world problems, generating parameters for zero-shot benchmarks with promising results (reflected in improvement over training data). ii) DnD's good performance on the 7B group, despite poor performance of training LoRAs (pass@k=0.0), stressing that *our method is not simply memorizing seen parameters, but learn to establish data-parameter mappings and adapt LLMs for novel datasets.*
**Math.** In this part, we report more results for math experiments in Table 14. Similar to common sense reasoning and coding tasks, DnD continues to work well on math datasets, improving largely over the average accuracy of training data on zero-shot benchmarks. This showcases its drag-and-drop ability has wide application scenarios. Also, results on Qwen2.5-7B prove its promising scalability.

| test set \ train set | IIO-Mini | ToT-Mini | Math-Plus | Mu-Math | Competition | Math-QA | Qwen | train set avg. | DnD |
|---|---|---|---|---|---|---|---|---|---|
| gsm8K | 68.6 | 35.4 | 68.3 | 22.8 | 31.0 | 31.4 | 64.4 | 42.9 | 66.3 |
| MATH | 30.0 | 1.5 | 30.2 | 7.2 | 16.5 | 3.5 | 29.3 | 14.8 | 23.9 |
| average accuracy improvement: **23.4** on gsm8K, **9.1** on MATH. | | | | | | | | | |

(a) Even with some low-performing training LoRAs (*i.e.*, accuracy less than 50% of base Qwen), DnD still maintains good zero-shot performance, showing the drag-and-drop ability to fit LLMs for novel datasets.

| test set \ train set | IIO-Mini | ToT-Mini | Math-Plus | Mu-Math | Competition | Math-QA | Qwen | train set avg. | DnD |
|---|---|---|---|---|---|---|---|---|---|
| gsm8K | 88.3 | 50.7 | 68.3 | 61.1 | 88.3 | 60.7 | 81.2 | 65.9 | 83.1 |
| average accuracy improvement: **17.2** on gsm8K. | | | | | | | | | |

(b) Using larger Qwen2.5-7B, DnD continues to drag-and-drop it, indicating our method has good scalability.

Table 14: DnD continues to work well on math tasks, showing our method has broad applicability.

## B.2 Inference Efficiency Analysis

In this part,we further analyze DnD's efficiency by presenting its memory usage and inference time for generating various LLMs on respective tasks. We show the cost of generating one single model on a NVIDIA A100 80G GPU in Table 15.

| metrics | common sense | math | coding | multimodal |
|---|---|---|---|---|
| time (second) | 0.11 | 0.53 (1.5B) 
 0.55 (7B) | 0.70 (1.5B) 
 0.73 (7B) | 0.61 |
| memory cost (GB) | 9.59 | 15.43 (1.5B) 
 16.22 (7B) | 19.17 (1.5B) 
 20.48 (7B) | 20.31 |

Table 15: Inference time and memory cost for different LLMs generation. All metrics are measured on a single NVIDIA A100 80G GPU. The time and memory is the cost to generate a single model.

## B.3 More Ablation Studies

**Can answer serve as condition in more complex scenarios?** In Section 3.4, we've explored different condition types and come up with the conclusion: *simple, identical answers will limit the diversity of conditions.* In this part, we explore using more complex answers from math datasets in Section 3.3 as conditions for building condition-parameter pairs and training DnD.

From results in Table 16, we can observe that more complex, diverse answers can lead to better performance. This may because their diversity is able to provide DnD with a comprehensive view of condition-parameter mapping. However, as answers are typically much longer than prompts in math and coding datasets due to problem complexity, we still recommend to use prompts as conditions for DnD.

| condition type | accuracy on gsm8K(%) |
|---|---|
| answer | 64.0 |
| **prompt** | **66.3** |

Table 16: Answer in math task can serve as conditions, but prompts still work better.

