# OpenReview forum: "Drag-and-Drop LLMs: Zero-Shot Prompt-to-Weights"
_NeurIPS.cc/2025/Conference — NeurIPS 2025 poster_

### Official Review · Reviewer_AWwr · 2025-07-01

**Clarity:** 3
**Significance:** 3
**Originality:** 4
**Rating:** 5
**Confidence:** 3

**Summary:**

The paper proposes DnD to replace PEFT. DnD takes unlabeled task prompts as input to generate LoRA parameters. The paper shows that the idea is working on many datasets and provides several ablations.

**Questions:**

I'll decrease score if non of the weaknesses is solved and raise to 6 if most of the weaknesses are solved.

**Ethical Concerns:**

["NO or VERY MINOR ethics concerns only"]

**Final Justification:**

As comment to the authors' rebuttal.

**Quality:**

3

**Strengths And Weaknesses:**

Strength:

1. It's crazy for me to see (if) the idea is working.

Weakness:

0. The tokenization in equation 4 is not clear. The author should formulate the tokenization out. (I had a glance on RPG, there are lots of implementation details. The author should detail the implementation used in this paper with formulation, possibly in appendix, rather than simply use tokenization[58] and force readers to go through RPG.)
1. It's not clear that what if different datasets use different settings of lora? Such as the low rank is different among different datasets.
2. Table 2 is not comprehensive, making the results unconvincing: the table only shows science as results of cross domain. What about others? The author should show many. It's fine if the results are bad. Bringing the whole picture is important.
3. For all experiments (let me know if I'm wrong), DnD is trained on only one domain. The paper should consider multiple domains for training DnD.
4. The paper misses an important result. What is the DnD's performance on dataset A, if DnD is trained on datasets A+B+C+D+...? So A is in the training data. This let us know the DnD's effect on training tasks.
5. It's not clear why in Figure 5, DnD performs better than RPG.
6. The setting of Figure 6 id not clear. What is DnD trained on? Multiple Fig. 6 are supposed be shown in different datasets.

[let me know if the any of my question is solved by appendix, I did not go through all the appendix]

---

> ### Author Rebuttal · Authors · 2025-07-29
>
> Thanks reviewer AWwr for constructive comments. We are delighted to discuss the raised questions below and sincerely hope these answers can address the concerns.
>
> > **Q1:** The tokenization in equation 4 is not clear. The author should formulate the tokenization out, possibly in appendix.
>
> **A1:** Thanks for the comment. We formulate the tokenization process as follows:
> - **1. Normalize each layer's weight of collected checkpoints:** Each layer's weight is normalized by its element-wise mean and std: $\mu = mean(W), \sigma= std(W), \hat{W} = \frac{W-\mu}{\sigma}$, where $W^{i}$ are the $i$ th layer's parameters.
> - **2. Slice model parameters of each layer into tokens of specified size:** We first slice model weights into equal-sized tokens: $\hat{W}^{i} = [ \hat{W}_1^i , \hat{W}_2^i , ... \hat{W}_n^i  ]$, where $\hat{W}_j^i$ is the $j$ th token of $i$ th layer.
> - **3. Stack all tokens together:** All tokens are stacked together to form the shape of **[N′,L′,C′]** mentioned in **line 146**.
> - **Shorter version:** For the convenience, we also present tokenization process in the form of **mind chains**: **weights normalization** -> **weights tokenization** -> **stack tokenized weights**.
> - **Action:** We follow reviewer AWwr's advice and take the following actions:
>     - We add the above formulations and details to **Section 2.5**, **line 143** in the revision.
>     - We sincerely apologize for causing obstacles in reading for the readers and we will carefully polish and constantly improve our work.
>
>
> > **Q2:**  It's not clear what if different datasets use different settings of LORA? Such as the low rank is different among different datasets.
>
> **A2:** Thanks for the comment.
>
> - DnD works well in parameter generation with **consistent structures**, e.g., LoRA adapters with the **same rank**.
> - Currently, DnD does not support generating LoRA adapters with **varying ranks**.
> - **Variable structure parameter generation** is a very promising and valuable insight, especially for tasks related to **model deployment**. Specifically, we can generate different parameters satisfying various **hardware requirements**.
> - To achieve this, we plan to conduct the following explorations:
>     - Devising **tokenization method** that supports variable structure parameter generation.
>     - Embedding this structural information as **condition** in generation process.
>     - Generating LoRA adapters that is **robust to pruning on ranks** may also be a solution.
>
> Once more, we thank reviewer AWwr for the constructive advice, and we are going to explore variable structure parameter generation in the future.
>
>
> > **Q3:** Table 2 only shows science as the cross-domain results. What about others?
>
> **A3:** Thanks for the comment.
>
> - Before answering the question, we show the **max prompt length** of different tasks as follows:
>
>     |task|max prompt length|
>     |-|-|
>     |common sense reasoning| 384 |
>     |science|416|
>     |coding| 26624 |
>     |math|4608|
>     |multimodal|1536 |
>
> - **Findings:**
>     - common sense reasoning and science dataset's max prompt length is **close**.
>     - common sense reasoning dataset's max prompt length **differs significantly** from math and coding datasets.
>
> - **Why can't we drag-and-drop common sense reasoning to the math/coding dataset?**
>     - **Overall Summarization:** The gigantic difference in max prompt length makes DnD trained on common sense reasoning hard to encode math/coding datasets' prompts, **losing information** for the given task.
>     - In **line 563**, **Appendix A.3** of the paper, we introduce that the model processes sequences of fixed length (512).
>     - Therefore, for **common sense reasoning**, we padding **384** to **512** as the sequence length dimension.
>     - **Math** and **coding** datasets' prompt length is **much larger** than 512. If we want to use math or code prompts to inspire DnD trained on common sense reasoning without introducing extra modules, we need to **average pool** or **randomly drop** **26624** or **4608** to 512.
>     - This operation is an adding-on to our method **(NOT in an end-to-end manner)** and **may lose much information** for the given task. Therefore, we fail to drag common sense reasoning to math or code domains.
>
> - **Why we can drag-and-drop common sense reasoning to science dataset**
>     - Max prompt length in **science dataset** is smaller than 512, so its information for the given task can be **well captured** in DnD's processing.
>
> - **Whole picture demonstration:**
>     - **Direct math-to-coding cross domain:** Similar to the above discussion, math/coding datasets' max prompt length also varies **(4608 vs 26624)**, compressing coding prompts to math's input length will also cause **information loss**.
>     - **Mixed domain training:** Train math and coding datasets **together** may be a solution for cross-domain dragging, as reviewer AWwr's **Question 4**. Please refer to our **Answer 4** for more details.
>
>
>
> > **Q4:** Train DnD on multiple domains.
>
> **A4:** Thanks for reviewer AWwr's comment. We conduct the experiment as follows:
>
> - **Setting:** We involve datasets and checkpoints **in math and coding tasks** to train DnD.
> - **Implementation Details:** Due to time limits, we directly apply BERT's padding to **math** prompt embeddings to ensure all embeddings are of **equal sequence length**. Simply speaking, **4608 + padding $\rightarrow$ 26624**.
> (Note: Due to dramatic differences **(4608 vs 26624)** in prompt lengths, we are not certain this is the best solution. We will continue exploring better solutions.)
> - **Results:**
>
>     |test set|avg of training LoRAs| DnD with joint trainig |$\Delta$|
>     |-|-|-|-|
>     |HumanEval (pass@10)|39.4|29.6|$\downarrow$ 9.8|
>     |gsm8k (accuracy)|46.3|65.3|$\uparrow$ 19.0|
>
> - **Findings and Conclusions:**
>     - Mixed domain training doesn't help DnD drag to tasks with longer prompt length.
>     - Mixed domain training enables DnD to cross domains with shorter prompt length.
>
> - **Action and Future Plans:**
>     - We add these results to **Section 3.5** and **Table 7** of the paper, with a paragraph named **Mixed domain training.**
>     - We will continue exploring mixed domain training with plans as below:
>         - Using **feature compression** methods to obtain sequences of **similar length**.
>         - Applying **data augmentation** methods instead of **padding**, since it may cause information loss.
>
> We are still improving DnD for the better, and thanks reviewer AWwr for offering the advice.
>
> > **Q5:** What is DnD's performance on dataset A, if DnD is trained on datasets A+B+C+D+...?
>
> **A5:** Thanks for the comment. We answer as follows:
> - We conduct this experiment in **Section 3.5**, **line 302**, paragraph **Comparison with previous methods**, and show the results in **Figure 5**. For the convenience, we summarize and put the results in the table below.
> - **Settings:** We train DnD on **all 6 common sense reasoning datasets except ARC-c** and evaluate on training datasets.
> - **Results:**
>
>     |dataset|training LoRAs|DnD|
>     |--|---|--|
>     |ARC-e|67|69|
>     |OBQA|44|51|
>     |PIQA|30|36|
>     |HellaSwag|51|51|
>     |WinoGrande|81|38|
>     |BoolQ|46|64|
>
> - **Findings and Conclusions:**
>     - DnD surpasses training LoRAs in most cases, revealing good ability also in **close-set generation**.
>
>
> > **Q6:** It's not clear why in Figure 5, DnD performs better than RPG.
>
> **A6:** Thanks for the comment. We conclude that DnD outperforms RPG for the following reasons:
>
>
>
> - **Conditioning Mechanism: data feature vs binary embedding**
>     - DnD directly utilizes **input prompts' embeddings** as conditions, which equips it with an understanding of the transition from data space to weight space.
>     - RPG uses only **binary embeddings** as conditions, which, though it succeeds on **simple tasks such as CIFAR-10**, fails in complex scenarios such as language tasks.
>     - In close-set training, datasets in common sense reasoning have **subtle differences**, but binary embedding is hard to serve as conditions for this fine-grained scenario. Therefore, it is more appropriate to use data prompts as conditions.
>     - Binary embedding is **discretely distributed** while prompt embeddings are **continuous in semantic space**. In open-set testing that requires generality, prompt embeddings as conditions can have better zero-shot ability.
>
>
> > **Q7:** The setting of Figure 6 is not clear. What is DnD trained on? Multiple Fig. 6 are supposed be shown in different datasets.
>
>
> **A7:** Thanks for this suggestion, and we will make the following adjustments to the paper.
>
> - **What is DnD trained on?** Training datasets are **ARC-e, OBQA, PIQA, HellaSwag, WinoGrande, BoolQ**. **Gray circles** contains parameters whose **distribution is close** after PCA.
>
> - **Add more visualization Figures:**
>     - We add coding, math, and multimodal weights' visualization Figure in revision's **Figure 7**, **Figure 8**, and **Figure 9**.
>     - Though NeurIPS 2025 **can't upload Figures** in rebuttal phase, we provide the **figure captions** here:
>     - **Coding caption:** In complex scenarios, DnD manages to drag-and-drop LLM weights towards the desired location with good performance.
>     - **Math caption:** DnD also works well on math datasets, showing robust generality and wide application scenarios.
>     - **Multimodal caption:** Our method continues to show its magic on multimodal tasks, indicating it has broad applicable potential without the limitation of modality.
>     - Moreover, we add visualization of **other common sense reasoning datasets** to **Appendix C**.
>
> We sincerely hope our response can address the concerns, and hope to hear more from reviewer AWwr.

---

> ### Author Response · Authors · 2025-08-04
> **Appreciate your feedback and welcome any further question**
>
> Dear Reviewer,
>
> We hope our rebuttal has addressed your concerns. To save your time, we make a summary of our rebuttal as follows:
>
> 1. We formulate the tokenization process and also put it in the revision.
>
> 2. We discuss DnD's current limitation on fixed structure generation and our plans on realizing it.
>
> 3. We answer the question by:
>     - Analyze different datasets' max prompt length and show that cross-domain dragging may lose information in some scenarios theoretically.
>     - Conduct experiments on complex cross-domain dragging (math and coding).
>
> 4. We conduct experiments of training on multiple domains.
>
> 5. We conduct experiments on close-set generation.
>
> 6. We discuss the reasons for DnD to surpass RPG.
>
> 7. We add more visualization results in the revision and show their captions.
>
> Looking forward to your reply!

---

> ### Comment · Reviewer_AWwr · 2025-08-05
>
> I appreciate the authors' answer to my questions. My questions are mostly answered, though many unsatisfied facts are there such as: A2: DnD currently cannot be applied to varied LoRA rank; A3: due to prompt length mismatch we cannot perform cross-domain; A5: very low performance of DnD on WinoGrande. I believe those are empirical facts which honestly reported by the authors.
>
> I increase my score to 5.

---

> > ### Author Response · Authors · 2025-08-05
> > **Thanks for your acknowledgement**
> >
> > Dear Reviewer,
> >
> > Thanks for your acknowledgement of our rebuttal. We remain honest about our limitations and are working on some of them currently.
> >
> > Variable structure generation is a highly promising and challenging direction, which takes endeavors to achieve. During our exploration, we find that doing **variable width** generation is simpler than **variable depth**, and it is a helpful finding.
> >
> > For cross-domain dragging, due to **computational resource limit**, the domain number we involve for training is small. We plan to incorporate as many domains as possible, in order to improve DnD's generality.
> >
> > We will state all limitations in the revision, and continue to address them in subsequent works.
> >
> > If you have any further questions, feel free to discuss them with us and we are more than happy to solve them. Thank you.

---

### Official Review · Reviewer_MX6i · 2025-07-02

**Clarity:** 3
**Significance:** 3
**Originality:** 3
**Rating:** 5
**Confidence:** 5

**Summary:**

The paper introduces Drag-and-Drop LLMs, a new way to quickly adapt large language models to new tasks without any extra training. Instead of fine-tuning a model for each task, DnD takes a few example prompts and directly generates the necessary weight updates using a lightweight encoder and a special decoder. It’s super efficient, up to 12,000 times faster than traditional methods, and still manages to beat or match performance on tasks like reasoning, math, coding, and even multimodal problems. Best of all, it works out-of-the-box on new tasks it hasn’t seen before.

**Questions:**

(1) How sensitive is DnD to the diversity and quality of the training prompt-checkpoint pairs?
(2) Why was a hyper-convolutional decoder chosen over more common alternatives, and how does it compare? I'm still a llitle bit confused why this architecture.
(3) What are the main failure cases or limitations of DnD in practice?

**Ethical Concerns:**

["NO or VERY MINOR ethics concerns only"]

**Limitations:**

Please see weakness and Questions. Several questions need to be further addressed.

**Quality:**

3

**Strengths And Weaknesses:**

This paper has some strengths:
(1) The method achieves dramatic reductions in adaptation overhead—up to 12,000× faster than traditional fine-tuning—making it highly practical.
(2) Well-written and easy to read.
(3) Extensive evaluations, ablation studies, scalability tests, and comparisons with both traditional and recent methods (e.g., RPG) build a solid case for the method’s robustness and utility.

This paper has some weaknesses:
(1) Several important implementation details are deferred to the appendix, which can interrupt understanding when reading the main paper.
(2) Since DnD is trained on LoRA outputs, it may not generalize well to tasks or models where LoRA performs poorly. How should I address this?

---

> ### Author Rebuttal · Authors · 2025-07-30
>
> Thanks reviewer MX6i for appreciating the efficiency, extensive evaluations, and novelty of our work. We also thanks the helpful suggestions and insightful questions offered by reviewer MX6i, we address these as follows.
>
>
> > **Q1:** Several details are deferred to the appendix, which can interrupt understanding of the main paper.
>
>
>
> **A1:** Thank for the comment, we make several **modifications** in the revision:
> - In **Section 2.5**, **line 135**, we put **Tabel 8** in **Appendix** here to delve into **hyper convolutional decoder's structure**.
> - In **Section 2.5**, **line 145** of the paper, we formulate the detailed **tokenization process** for better clarity.
> - In **Section 3.1**, **line 162** of the paper, we put **Table 7**, **Table 9** in **Appendix** here, to give a thorough introduction of **our training setting** and **checkpoint collection**.
>
> - **Conclusion:**
>     - Following reviewer MX6i's suggestion, we will add more details in the revision.
>     - We apologize for interrupting the understanding of readers, and we will continue to polish our work.
>
> > **Q2:** How should we apply DnD to scenarios where LoRA performs poorly?
>
> **A2:** Thanks for the comment. We admit that LoRA may perform poorly and not be as good as foundation LLMs in some scenarios. We clarify our method's characteristics:
> - Our work mainly explores **LoRA scenarios in language tasks**. However, our method is **independent** of model structures, which means we can also generate **entire** foundation LLMs.
> - By arithmetic estimation, we can generate full parameters of Qwen2.5-**0.5B** on **8 H200 GPUs**. But we are currently **limited by resources** to generate foundation LLMs of larger sizes, e.g., 7B or larger.
> - We will continue to explore **scaling up** of parameter generation, broadening the application scenario of this technology.
> - reviewer MX6i's question can also be interpreted as DnD's **robustness** after training on **low-quality** LoAR adapters (Please feel free to correct us if there is any misunderstanding).
> - In some scenarios, we may only have **worse-performing** LoRAs for training, and our method still shows strong **data-adapting** ability. Please kindly refer to **A3** for more details.
>
>
> > **Q3:** How sensitive is DnD to the diversity and quality of the training prompt-checkpoint pairs?
>
>
> **A3:** Thanks for the question. We answer as follows:
>
> - **Diversity:** We conduct ablation study of **datasets arrangement** in **line 233** and **Table 4c** of the paper. For convenience, we put the results here.
>     - **Results:**
>         |dataset arrangement | $\Delta$ with training LoRAs|
>         |-|-|
>         | 6 $\in$ train,1 $\in$ test |$\uparrow$ 12.1 |
>         | 4 $\in$ train, 3 $\in$ test |$\uparrow$ 11.4 |
>         | 3 $\in$ train, 4 $\in$ test |$\uparrow$ 0.8 |
>         | 2 $\in$ train, 5 $\in$ test |$\downarrow$ 1.4 |
>     - **Conclusions:**
>         - Generally, **more** training datasets lead to **better** performance improvement.
>         - As datasets used for training **lessen**, the average improvement of DnD **drops accordingly**.
>         - Therefore, **basic amount** of training samples is needed for DnD to learn condition-parameter correlations.
>
> - **Quality:** Thanks reviewer MX6i for the question, and we conduct more ablation studies as follows.
>     - **Setting:** In addition to collecting **well-trained** checkpoints, we collect checkpoints **at the beginning** of training trajectory, i.e., checkpoints **not converged** on the training sets.
>     - **Results:** (training LoRAs are trained **ARC-e, OBQA, PIQA, HellaSwag, WinoGrande, BoolQ** and test set is **ARC-c**)
>
>         |checkpoint quality|avg of training LoRAs|DnD generated LoRA|improvement|
>         |-|-|-|-|
>         |w/o converged|27.3|46.3| $\uparrow$ 19.0 |
>         |converged|37.5|51.6| $\uparrow$ 14.1 |
>
>     - **Findings:**
>         - DnD **maintains** good performance even with **worse-performing** LoRAs.
>         - DnD's performance **drops** with training LoRAs that are **insufficiently tuned**.
>         - Even DnD tuned on worse LoRAs surpasses **converged** training LoRAs on **zero-shot** test set.
>
>
>     - **Conclusions:**
>         - DnD works well with worse training LoRAs, which reveals that our method has strong **data adapting** ability.
>         - Training on sub-optimal LoRA can affect DnD's performance, since it isn't aware of **good weights' distribution**.
>
>
>     - **Action:** We add these results and analysis to **Section 3.4**, **Table 4e** in the revision.
>
> > **Q4:** Why was a hyper-convolutional decoder chosen over more common alternatives?
>
>
>
> **A4:** Thank for the comment, we answer as follows.
> - According to previous works [4,6,7,8,9], we can conclude that common **backbones** in parameter generation can be:
>     - **Pure latent diffusion[3]:** [4,6,7]
>     - **Conditional latent diffusion:** [6]
>     - **Mamba [5] + latent diffusion:** [9]
>     - **LSTM [1] + latent diffusion:** [9]'s ablation
>     - **Casual transformer[2] + latent diffusion:** [9]'s ablation
> - We would like to summarize different backbones' properties as follows:
>     | backbone | large scale generation | single step generation | conditional generation |
>     |-|-|-|-|
>     |latent diffusion| $\times$ | $\times$ | $\times$ |
>     |conditional latent diffusion| $\times$ | $\times$ | $\checkmark$ **(text discriptions)**|
>     |RPG with Mamba|$\checkmark$ **(up to 200M)** | $\times$ | $\checkmark$ **(binary embeddings)**|
>     |RPG with LSTM|$\checkmark$ **(up to 200M)** | $\times$ **(slower than Mamba)**| $\checkmark$ **(binary embeddings)**|
>     |RPG with causal Transformer| $\checkmark$ **(up to 200M)**| $\times$ **(slower than Mamba)**| $\checkmark$ **(binary embeddings)**|
>     |**Hyper convolution (ours)**| $\checkmark$ **(up to 0.5B)**| $\checkmark$ **(0.1s)**| $\checkmark$ **(prompts in data samples)**|
>     (Generation parameters are tested on **8 H200 GPU**.)
>     - **Findings:**
>         - **Pure** latent diffusion models can not generate large models.
>         - **Mamba, LSTM, and transformer** can generate parameters of larger scale **(up to 200M)**, but this manner of generation **lacks efficiency**.
>         - Hyper convolution can achieve both large scale **(up to 0.5B)** and conditional **(data as conditions)**. Moreover, it can achieve **real-time** generation with only **0.1s**.
> - **Broader impact of real-time parameter generation:**
>     - **Fast Domain Adaptation**
>     - **Customizing Models for Users**
>     - **Others**
>
> **References:**
>
> [1] Hochreiter, Sepp. et al. Long short-term memory, MIT press 1997
>
> [2] Vaswani, Ashish. et al. Attention is all you need, NeurIPS 2017
>
> [3] Rombach, Robin. et al. High-resolution image synthesis with latent diffusion models, CVPR 2022.
>
> [4] Wang Kai. et al. Neural network diffusion, arXiv 2024.
>
> [5] Gu, Albert. et al. Mamba: Linear-Time Sequence Modeling with Selective State Spaces, COLM 2024
>
> [6] Jin, Xiaolong. et al. Conditional lora parameter generation. arXiv 2024
>
> [7] Soro, Bedionita. et al. Diffusion-based neural network weights generation, arXiv 2024.
>
> [8] Li, Zexi. et al. Text-to-model: Text-conditioned neural network diffusion for train-once-for-all personalization, arXiv 2024
>
> [9] Wang Kai. et al. Recurrent diffusion for large-scale parameter generation, arXiv 2025.
>
>
> > **Q5:** What are the main failure cases or limitations of DnD in practice?
>
>
> **A5:** Thank for the comment. DnD's current limitations are:
>
> - **Fixed structure generation:** Currently, DnD focuses on generating models with the same structure for different tasks. It currently doesn't support generating models of **different structures**. We realize that **variable structure generation** is a crucial and promising direction, and we will explore it in the future.
> - **Cross Domain Generalization:** Though we successfully drag common sense reasoning to science dataset in **Table 2** of the paper, crossing domains with larger gap remains challenging, as suggested in our **A1 to reviewer BsLp**.
>     - **Potential Solution:** We are going to incorporate a large amount of diverse training LoRAs (as many as possible), aiming at empowering DnD with stronger cross-domain ability.
>
> We add these limitations and relevant analysis to the revision. We appreciate reviewer MX6i's opinion and hope to discuss more.

---

> ### Author Response · Authors · 2025-08-04
> **Appreciate your feedback and welcome any further question**
>
> Dear Reviewer,
>
> We hope our rebuttal has addressed your concerns. To save your time, we make a summary of our rebuttal as follows:
>
> 1. We add multiple important details to the main part in the revision.
>
> 2. We discuss DnD's potential of whole foundation LLM generation and our plans in realizing it.
>
> 3. We conduct experiments to explore collected checkpoints' diversity and quality on DnD's performance.
>
> 4. We list a table to discuss various common backbones' advantages and disadvantages, showing our design motivation of DnD.
>
> 5. We discuss DnD's current limitation of fixed structure generation.
>
> Looking forward to your reply!

---

> ### Comment · Area_Chair_RzQE · 2025-08-05
> **Resond to authors ASAP**
>
> Dear Reviewer MX6i,
>
> Please read the authors' rebuttal and other reviewers' comments as early as possible. You are encouraged to provide your further feedback and engage in a discussion with the authors.
>
> AC

---

> ### Author Response · Authors · 2025-08-07
> **Looking forward to hear from reviewer MX6i**
>
> Dear reviewer MX6i,
> Thanks for your acknowledgement of our work and the efforts made in reviewing it. As the rebuttal phase is near its end, we would like to know if our rebuttal has addressed your concerns. For convenience, we summarize our responses and provide a **shorter version** here.
> > **Q1:** Implementation details deferred to appendix.
>
> **A1:** We adjust the content and put more details in the revision. We specify the details in our rebuttal.
> > **Q2:** How can DnD apply to scenarios where LoRA performs poorly?
>
> **A2:** We discuss DnD's potential of **generating whole parameters** of foundation LLMs, and DnD's robustness of **training on worse-performing LoRAs**.
> > **Q3:** Training LoRAs' diversity and quality on DnD.
>
> **A3:** We **conduct experiments** of training LoRAs' quality and diversity, together with analysis of empirical findings.
> > **Q4:** Why choose a hyper-convolutional decoder?
>
> **A4:** We list a **table** of common backbones in parameter generation to analyze their strenghs and weaknesses, and further clarify our motivation of choosing hyper convolutional decoder.
>
> > **Q5:** What is the limitation of DnD?
>
> **A5:** We discuss DnD's limitation of fixed structure generation, and provide our future plans on achieving variable structure generation.
>
> Thanks again for your time and efforts in reviewing our work and we are looking forward to hear from reviewer MX6i.
> Authors

---

### Official Review · Reviewer_BsLp · 2025-07-02

**Clarity:** 2
**Significance:** 2
**Originality:** 3
**Rating:** 5
**Confidence:** 3

**Summary:**

This work proposes a method to generate LoRA adapter weights conditioning on task-relevant prompts. The weight generator is trained from pairs of prompt and finetuned LoRA weights. Experiments demonstrate the proposed method can achieve better zero-shot accuracy on unseen task and cross-domain task compared to other trained LoRA adapters

**Questions:**

1.	In line 169, how is “the average accuracy of training LoRAs” calculated? Suppose the test set is ARC-e, does “training LoRAs” correspond to each LoRA adapter trained from the other tasks (OBQA, ARC-c …) respectively? If so, why not compare DnD with one LoRA adapter finetuned on all other test sets? Since the training sample of DnD generator includes adapters parameter produced from all other test sets.
2.	In Table 3, why is the improvement of DnD in Multimodal task minor compared to other tasks?
3.	In practical deployment, does DnD require sufficient number of prompts (which is determined by the length of prompt batch hyperparameter during training) to start generating the weights for a task?

**Ethical Concerns:**

["NO or VERY MINOR ethics concerns only"]

**Final Justification:**

The authors have well addressed my concerns. I thus increase the score to 5.

**Limitations:**

yes

**Paper Formatting Concerns:**

No formatting concerns.

**Quality:**

3

**Strengths And Weaknesses:**

## Strengths
1.	The idea of generating LoRA weights using task prompt is interesting and novel.
2.	Considerable performance boost in the zero-shot accuracy in various tasks compared to other trained LoRA adapters.
3.	Extensive evaluations on key design factors of the proposed method.

## Weakness
1.	The author did not explain why DnD has good generalization even in cross-domain task. In Table 2, the DnD generated weights even outperforms the LoRA adapter trained on the target domain, which appears counter-intuitive given DnD is never trained on adapters related to science-dataset.
2.	The comparison of DnD’s performance with foundation LLMs should be included in Table 1-3 instead of conducting a separate ablation study. It seems that most of the training LoRAs have a degraded task accuracy than the base LLM. Therefore, the actual performance gain brought by DnD may be exaggerated.
3.	The emphasis on 12000x speedup of DnD compared to full-shot tuning does not look reasonable to me. The building of DnD dataset already involves training a number of LoRA adapters for extended iterations, which takes much longer than training one specialized LoRA adapter.

---

> ### Author Rebuttal · Authors · 2025-07-29
>
> Thanks reviewer BsLp for acknowledging the novelty, performance gains and extensive evaluations of our work. We answer reviewer BsLp's questions below.
>
> > **Q1:** Why does DnD outperform those LoRAs tuned on the science dataset?
>
> **A1:** Thanks for this question.
> - **Clarification:**
>     - In **Table 2**, the training LoRAs are **trained on ARC-e, OBQA, PIQA, HellaSwag, WinoGrande, BoolQ (common sense reasoning datasets, NOT science datasets)**.
>     - These training LoRAs are **never trained on science dataset**.
> - **Action:**
>     - We add **"Note training LoRAs are trained on ARC-e, OBQA, PIQA, HellaSwag, WinoGrande, BoolQ, and they've never seen a science dataset."** in **line 174**, paragraph **Cross-domain Drag-and-Drop.** for better clarity.
>
> > **Q2:** Why does DnD succeed in the cross-domain (source $\rightarrow$ target) dragging?
>
> **A2:** We answer as follows.
> - DnD learns the **data-to-weight mapping**, instead of **fitting its parameters for a given dataset** like traditional LoRA tuning.
> - Moreover, DnD learns a parameter generator that is **data-adapting**.
> - Therefore, in cross-domain scenarios, it utilizes target domain data as **inspiration** and generate respective parameters, which surpasses source domain tuned LoRA in terms of **generality**.
> - For convenience, we summarize the differences between DnD and traditional LoRA tuning below:
>
>     |property|LoRA tuning|DnD|
>     |-|-|-|
>     | objective | fitting data in the source domain | learning data-to-weight mapping |
>     |origin of generality | domain gap | domain gap+target data inspiration |
>
> - The generality of our method also depends on the domain gap, and since the science dataset is **similar** to common sense reasoning, DnD is easier to generalize.
>
> - To explore the domain gap's influence, we test cross-domain with a larger domain gap, i.e., **coding-to-math**. (More in **A4 of reviewer AWwr**.)
>     - **Setting:** We use datasets and checkpoints in **math and coding tasks** to train DnD jointly.
>     - **Results:**
>         |test set|avg of training LoRAs| DnD with joint trainig |
>         |---|----|-----|
>         |gsm8k (accuracy)|46.3|65.3|
>     - **Conclusion:**
>         - DnD improves largely over its training LoRAs on gsm8k, showing it has the potential of **cross complex domains: from coding to math**.
>     - **Action:** We add these results to **Section 3.5** and **Table 7** in the revision.
>
>
>
> > **Q3:** The comparison of DnD with foundation LLMs should be included in Table 1-3.
>
> **A3:** Thanks for this advice, we change **Table 1-3** as follows.
>
> - **Modification:**
>     - We add the results in **Table 5** to **Table 1-3**.
>     - The relevant analyses are also moved to the respective sections in the revision.
> - **Performance Comparison among different methods:**
>     - Overall, DnD achieves the best performance among training LoRAs and base LLMs.
>     - Despite being trained on **worse-performing** LoRAs compared to base LLMs, DnD still obtains **better performance** than base LLMs on target tasks, showcasing our method has **strong perception and adaptation** of target datasets.
>
> > **Q4:** The emphasis on 12000x speedup of DnD compared to full-shot tuning does not look reasonable since the building of the DnD dataset already involves training LoRA adapters.
>
> **A4:** Thanks for the comment. We answer it as follows:
> - Training LoRAs on **zero-shot** test sets performance may be sub-optimal, and a practical solution is to use **full shot tuning** to get **dataset-specific** LoRA.
> - Our method, on the contrary, is a **once-for-all** parameter generator, that can adapt to other **novel** tasks unseen in training with proper training.
> - Therefore, the cost spent on DnD's training set building is **not targeted at the test set**. For test set adaptation, DnD takes **only one single forward pass**.
> - We appreciate the suggestion and make the following **modifications**:
>      - In **line 10** of the paper, "up to 12,000× lower overhead than full fine-tuning" $\rightarrow$ "up to 12,000× lower novel task adaptation cost once finished training".
>      - In **Figure 4b** of the paper, "comparable to full-shot tuning while being 12000× faster." $\rightarrow$ "performance comparable to full-shot tuning while saving 12000× adaptation cost after training".
>      - In **line 278** of the paper, "while being 12000× faster" $\rightarrow$ "being 12000× faster in adaptation once trained".
>
> > **Q5:** How is “the average accuracy of training LoRAs” calculated? Why not compare DnD with one LoRA adapter finetuned on all other test sets?
>
> **A5:** Thanks for the comment. We answer as follows:
>
> - **Calculation of avg acc**: The assumption is correct, “training LoRAs” are LoRAs trained on other tasks, and average accuracy is calculated via:  $\text{avg acc}=\frac{ \sum_i^n \text{acc}_i }{ \sum_i^n i }$, where $i$ starts from 1.
> - DnD takes prompt-checkpoint pairs to transform data to weights, i.e., **pairing** checkpoints with prompts they are **trained on**.
> - Training one type of LoRA adapter on all datasets will lead to prompts-checkpoints pairs being prompts from **various datasets** paired with **only one type of checkpoints**.
> - Trained on this data will instruct the generator to transform **all kinds of prompts** to **only one kind of weights**, limiting its generality for novel tasks.
> - To ensure **the number of datasets** used to train DnD and LoRAs is consistent, and considering the tight time of rebuttal, we design the experiment as follows:
>     - **Setting:** We train DnD on **ONLY one dataset PIQA** and use prompts from **one unseen test set ARC-c** to inspire the generator. **This guarantees the number of datasets used to train DnD and LoRAs is exactly the same**.
>     - **Results:** We report **accuracy on ARC-c**.
>
>         |training LoRAs (PIQA)|DnD generated LoRA|improvement |LoRA tuned on ARC-c (upper bound)|
>         |-|-|-|-|
>         |19.1|28.8| $\uparrow$ 9.7 |55.1|
>
>         Experiment results on **multimodal tasks** can also address the concern: (Note that Math-Vision and Math-Vista are zero-shot benchmarks that **don't contain a training set**, so there are no full-shot tuning results.)
>
>         |benchmark|training LoRAs|DnD generated|improvement|
>         |-|-|-|-|
>         |Math-Vision|23.0|24.3|$\uparrow$ 1.3|
>         |Math-Vista|61.5|62.3|$\uparrow$  0.8|
>
>     - **Findings:**
>         - With **same** number of training datasets, DnD obtains better **(9.7 improvement in accuracy)** performance than training LoRAs on novel test set.
>         - DnD trained on **only one dataset** is hard to generate competing parameters as full-shot tuning for novel datasets in inference.
>         - On multimodal tasks, DnD obtains better performance with training LoRAs with **minor improvement**.
>     - **Conclusion:**
>         - DnD's surpassing training LoRA indicates it works even with **limited knowledge** of data-to-weight transition.
>         - Using **only one kind** of checkpoints can't give DnD enough information about data-to-weight mapping, which limits DnD's generality.
>         - Results on multimodal tasks also prove that only one kind of dataset limits DnD generality to zero-shot benchmarks.
>     - **Action:** We add results and more experiment results (ongoing) in **Appendix B.3** in the revision.
>
> > **Q6:** Why is the improvement of DnD in the Multimodal task minor compared to other tasks?
>
> **A6:** Thanks for the comment. We answer it as below:
> - As discussed in **A4**, the number of training datasets is crucial in developing DnD's zero-shot ability.
> - In **line 164** of the paper, we use only **MathV360K** for multimodal task, due to **computation resources limit**. For example, CoMM takes up **2 TBs** of space and has more than**1B tokens** for training.
> - This limits the **diversity** of training LoRAs and makes DnD relatively **hard** to generalize well, so the improvement is minor.
> - **Plans:** We anticipate that the performance of the multimodal task can be improved via:
>     - incorporating **more datasets** in DnD's training, thus increasing the type of prompt-checkpoint pairs.
>     - **manually split MathV360K** into subsets (according to chartQA, mapQA, geometry, and other categories), collect LoRAs **tuned on each subset**, and train DnD on them.
>
> Thanks again for the comment. We would like to discuss more with reviewer BsLp.
>
>
> > **Q7:** In practical deployment, does DnD require a sufficient number of prompts?
>
>
> **A7:** Thanks for the comment, we answer this question as follows.
>
>
> - **Experiment:** We conduct more ablation studies on the **length of prompt batch**.
>     - **Setting:** We shrink the length of prompt batch to 8, 16, 32, 64 and evaluate their performance on **ARC-c**.
>     - **Results:**
>
>         |length of prompt batch|avg of training LoRAs|DnD|$\Delta$ with training LoRAs|
>         |-|-|-|-|
>         |8|37.5|33.2|$\downarrow$ 4.3|
>         |16|37.5|45.4| $\uparrow$ 7.9|
>         |32|37.5|47.3| $\uparrow$ 9.8|
>         |64|37.5|48.2| $\uparrow$ 10.7|
>         |128|37.5|51.6| $\uparrow$ 14.1 |
>         |512|37.5|52.0| $\uparrow$ 14.5 |
>
>     - **Findings:**
>         - When **prompt batch** is **too small (8)**, DnD's performance is not good.
>         - DnD can **surpass** training LoRAs with small prompt batch, i.e., **16**.
>         - As prompt batch grows **larger**, DnD's performance **improves** accordingly.
>
>     - **Conclusion:**
>         - DnD can achieve good performance with a small prompt batch, such as 16, showcasing our method's potential in **data shortage** scenarios.
>         - As the prompt batch **grows larger**, DnD obtains a better understanding of the given tasks (i.e., better likelihood estimation), reflected in DnD's **performance improvement**. We will further enlarge the prompt batch to obtain better performance in the future.
>
>     - **Action:** We add these results to **Table 4d** in **Section 3.4** to consolidate our work.
> We thank reviewer BsLp again for the comments.

---

> > ### Comment · Reviewer_BsLp · 2025-08-06
> >
> > I thank the authors for their detailed response and additional results. My concerns have been well addressed.

---

> > > ### Author Response · Authors · 2025-08-06
> > > **Thanks for reviewer BsLp's support and efforts in reviewing**
> > >
> > > We would like to thank reviewer BsLp's approval of our work and the efforts made during reviewing. Please let us know if you have any further questions. We are more than happy to address them and continue to improve our work.
> > >
> > > Authors

---

> ### Author Response · Authors · 2025-08-04
> **Appreciate your feedback and welcome any further question**
>
> Dear Reviewer,
>
> We hope our rebuttal has addressed your concerns. To save your time, we make a summary of our rebuttal as follows:
>
> 1. We clarify the datasets used for training LoRAs in Table 2.
>
> 2. We explore DnD's ability of cross-domain dragging by:
>     - List a Table of its differences with LoRA tuning.
>     - Conduct more cross-domain dragging experiment (coding to math).
>
> 3. We move the results of foundation LLMs to Table 1-3.
>
> 4. We explain the reason of stating the "12000* speedup" and rephrase our statement in the paper.
>
> 5. We answer the question by:
>     - Formulating how average accuracy is calculated.
>     - Conduct an experiment to show that the diversity of prompt-checkpoint pairs matters in DnD's performance.
>     - Experiments on multimodal tasks also support the conclusion.
>
> 6. We clarify that the minor improvement on multimodal tasks is due to the lack of diversity of prompt-checkpoint pairs, i.e., using only one dataset and LoRAs tuned on it.
>
> 7. We conduct ablation study on the number of prompts and analyze the results.
>
> Looking forward to your reply!

---

> ### Comment · Area_Chair_RzQE · 2025-08-05
> **Resond to authors ASAP**
>
> Dear Reviewer BsLp,
>
> Please read the authors' rebuttal and other reviewers' comments as early as possible. You are encouraged to provide your further feedback and engage in a discussion with the authors.
>
> AC

---

### Official Review · Reviewer_2zUf · 2025-07-03

**Clarity:** 2
**Significance:** 3
**Originality:** 2
**Rating:** 4
**Confidence:** 3

**Summary:**

This paper introduces Drag-and-Drop LLMs, a hypernetwork approach where prompt-conditioned generator maps a small batch of unlabeled task prompts directly to LoRA weight updates. Experiments on several text-based tasks show that DnD outperforms existing PEFT methods.

**Questions:**

Please see weaknesses

**Ethical Concerns:**

["NO or VERY MINOR ethics concerns only"]

**Final Justification:**

After reading through the discussion, I am deciding to maintain my score.

**Limitations:**

yes

**Quality:**

3

**Strengths And Weaknesses:**

- The motivation is clear: the proposed framework eliminates the need for per-task optimization while also sharing information across tasks via the parameter generator. The idea is, to my knowledge, novel. The efficiency gains on novel text tasks is very compelling.
- The paper provides several insightful ablations: in-domain vs cross-domain, conditioning, extractor, train/test split.
- Experimental results show that DnD consistently outperforms LoRA training
  -  For this experiment, I'd imagine the conclusion has a strong dependence on dataset size. For a representative set of datasets, could you report how LoRA performance scales with dataset size (while comparing to DnD numbers)?
- The learning objective was somewhat counterintuitive to me. You minimize an MSE loss to the fine-tuned LoRA updates, which would make sense when the set of good LoRA weights for a given task is unimodal, i.e. if you trained multiple times with different random seeds, the resulting weights would be close together. Is this something you see empirically? If not, why does the MSE loss work?

---

> ### Author Rebuttal · Authors · 2025-07-29
>
> Thanks reviewer 2zUf for recognizing our work's novelty and efficiency, and for raising insightful comments. We address the comments one by one as follows.
>
> > **Q1:** How LoRA performance scales with dataset size (while comparing to DnD numbers)?
>
> **A1:** While we are not completely certain about the specific dataset size that Reviewer 2zUf refers to and would appreciate a clarification, we did our best to provide as comprehensive an answer as possible based on our understanding.
>
> - **Dataset size used for training LoRAs:** We think reviewer 2zUf most likely refers to the dataset size used to train LLM LoRAs, that is, the **text dataset size**. Based on this, we perform an experiment to explore **dataset size**'s influence on **training LoRAs** and **DnD**.
>
>     - **Setting:** In our paper, we use the **entire dataset** to train and collect LoRA checkpoints. To ablate the influence of **dataset size**, we use **10\% of each dataset** to train LoRAs, then adopt these LoRAs for DnD's training, and evaluate their performance. Due to limited time, we only report test results on **ARC-c** (training LoRAs are obtained from ARC-e, OBQA, PIQA, WinoGrande, HellaSwag, and BoolQ).
>     - **Results:**
>
>         |dataset size|avg of training LoRAs|DnD|
>         |-|-|-|
>         |full (our paper)|37.5|51.6|
>         |10\%|29.8|49.1|
>
>
>     - **Findings:**
>         - Even trained with only 10\% data, DnD still maintains **good performance** (51.6\% $\rightarrow$ 49.1\%).
>         - As expected, training LoRAs' performance drops largely (37.5\% $\rightarrow$ 29.8\%) with only 10\% training data.
>         - DnD' performance drop is much smaller than training LoRAs on **novel** dataset (i.e., ARC-C)
>
>     - **Conclusions:**
>         - DnD is **less sensitive** to dataset size than **traditional LoRA tuning**.
>         - Despite the performance **drop** in training LoRAs, DnD **maintains** good performance. This indicates our method has **strong dataset adapting ability.**
>
>     - **Action:** We've added these results to **Table 4f** in **section 3.4** of the paper.
>
> - **Data sample number used to train LoRAs and inspire DnD respectively:** **Dataset size** may also refer to **the data** used to train LoRAs and inspire DnD. Take common sense reasoning as example, in **Table 7** and **Table 9** of the paper, LoRAs is trained on **5000 sample** for each task, while DnD accepts **128 prompts** to generate a LoRA adapter.
>     - We put respective data in the **table below** for the convenience:
>
>         |samples used for training LoRAs (DnD training data) | length of prompt batch (DnD testing)|
>         |--|--|
>         |**entire** dataset | **batch** of dataset, default: 128/1000|
>
>     - **Conclusion:** DnD uses very **few samples** as conditions to generate parameters, yet still obtains superior performance compared to training LoRAs (51.6\% vs 37.5\%).
>
>
> - **Dataset number used to train DnD:** reviewer 2zUf may refer to the **number of datasets (prompt-checkpoint pairs)** used to train DnD. In **Table 4c**, we conduct an ablation studies named **datasets arrangement** to explore, we put the table here for the convenience, and hope this can address your concern.
>     - **Table:**
>
>         |datasets arrangement|$\Delta$ with training LoRAs|
>         |--|--|
>         |6 $\in$ train, 1 $\in$ test|  $\uparrow$ 12.1 |
>         |4 $\in$ train, 3 $\in$ test|  $\uparrow$ 11.4 |
>         |3 $\in$ train, 4 $\in$ test|  $\uparrow$ 0.8 |
>         |2 $\in$ train, 5 $\in$ test|  $\downarrow$ 1.4 |
>
>     - **Conclusions:**
>         - Generally, **more** training datasets lead to **better** performance improvement.
>         - As datasets used for training **lessen**, the average improvement of DnD **drops accordingly**.
>         - Therefore, **basic amount** of training samples are needed for DnD to learn condition-parameter correlations.
>         - more datasets $\rightarrow$ more prompt-checkpoint pairs $\rightarrow$ deeper understanding of **data-to-weight transition** $\rightarrow$ **better generality**
>
>
>
> - **Length of prompt batch:** **Dataset size** may also refer to the **number of prompts used as inspiration** (length of prompt batch, default as 128) introduced in **line 139** of the paper, and we conduct more ablation studies accordingly.
>
>     - **Setting:** We both shrink and enlarge the length of prompt batch, turning it to 8, 16, 32, 64, 512 and explore their influence on DnD. Due to limited time, we only test on **ARC-c**.
>
>
>     - **Results:**
>
>         |length of prompt batch|avg of training LoRAs|DnD|$\Delta$ with training LoRAs|
>         |-|-|-|-|
>         |8|37.5|33.2|$\downarrow$ 4.3|
>         |16|37.5|45.4| $\uparrow$ 7.9|
>         |32|37.5|47.3| $\uparrow$ 9.8|
>         |64|37.5|48.2| $\uparrow$ 10.7|
>         |128|37.5|51.6| $\uparrow$ 14.1 |
>         |512|37.5|52.0| $\uparrow$ 14.5 |
>
>     - **Findings:**
>         - When **prompt batch** is **too small (8)**, DnD's performance is not good.
>         - DnD can **surpass** training LoRAs with small prompt batch, i.e., **16**.
>         - As prompt batch grows **larger**, DnD's performance **improves** accordingly.
>
>     - **Conclusion:**
>         - DnD can achieve good performance with small prompt batch such as 16, showcasing our method's potential in **data shortage** scenarios.
>         - As prompt batch grows **larger**, DnD obtains better understanding of the given tasks (i.e., better likelihood estimation), reflected in DnD's performance **improvement**. We will further enlarge the prompt batch to obtain better performance in the future.
>
>     - **Action:** We add these results to **Table 4d** in **Section 3.4** to consolidate our work.
>
> - **Number of collected checkpoints for each dataset:** The number of checkpoints used in each dataset may also be the answer.
>     - It can be conclude that **too few** checkpoints fails to learn the data-to-weight transitions for each dataset.
>     - While **too much** checkpoint can induce **extra training cost**. We use the settings below that have passed our **empirical examination.**
>
>     - **Setting:** Note that we use one dataset for **multimodal**, so the number of checkpoints is large.
>
>         | task | number of checkpoints |
>         |----|--------|
>         |common sense reasoning| 50 |
>         |coding/math 1.5B, math 7B| 80 |
>         |coding 7B| 100 |
>         |multimodal 3B|200|
>     - **Action:** We add these settings to **Table 9** in revision for better clarity.
>
> - **Conclusion:**
>     - We summarize our understanding of Reviewer 2zUf's question, and hope to have addressed their concern. We would appreciate a clarification and will be glad to provide further responses.
>
> >**Q2:** The design choice of MSE loss.
>
> **A2:**
>
>
> - **Checkpoint collection:** We collect checkpoints **for a given task** by:
>     - Performing **iterative LoRA fine-tuning** on one foundation model for specified epochs.
>     - Collecting checkpoints **at the end** of training trajectory.
>
> - **Distrubtion of LoRA adapters:** We agree with Reviewer 2zUf that **MSE makes sense when LoRA weights for a given task is unimodal**. So we conduct the following experiment to clarify:
>     - **Setting:** We compute the **average L2 distance** between parameters trained on the **same dataset** and those trained on **different datasets**. Additionally, we perform **Hartigan’s Dip Test** to assess unimodality, and report the **proportion of parameter elements** with **p-value > 0.05**. Due to time limits, we only conduct experiment on **ARC-c**.
>
>     |avg L2 distance (within ARC-c) |avg L2 distance (with other 6 common sense reasoning datasets)| ratio of p-value > 0.05 (dip test)|
>     |--|--|--|
>     | 4.0 e-09 | 2.2 e-06 | 0.865 |
>
>
>     - **Findings:**
>         - **L2 distance** within the same dataset is much **smaller (with magnitude of 500)** than with other datasets.
>         - **Most parameters (more than 85 percent)** of collected checkpoints conform to distribution with **p-value>0.05**, meaning that we can **accept the unimodal hypothesis**.
>
>
>
> - **Why we collect checkpoints only from one training trajectory for each dataset?**
>     - **Target:** Obtaining model parameters that perform well.
>     - **Single or multiple parameter patterns:** Based on our target, we foucs more on  developing mapping from data to single parameter pattern rather than decoding multiple patterns from given prompts.
>     - **Optimization difficulty consideration:** Mapping to single pattern is much easier to optimize. Since checkpoints collected from **multiple trajectory** is **not simple unimodal distribution**, making optimization much harder.
>
> - **Generating various parameter patterns is challenging but interesting.**
>     - We are aware that generating multiple high-performing and diverse parameters is difficult just as reviewer 2zUf mentioned.
>     - **Potential Solution:** It's possible to devise a **conditioning mechanism** that encodes **different random seeds'** or **network symmetries'** information in parameter generator's training.
>
> We appreciate the comments from reviewer 2zUf and hope the answers have addressed the concerns.

---

> ### Author Response · Authors · 2025-08-04
> **Appreciate your feedback and welcome any further question**
>
> Dear Reviewer,
>
> We hope our rebuttal has addressed your concerns. To save your time, we make a summary of our rebuttal as follows:
>
> 1. We consider dataset size from 5 different angles and answer them respectively.
>
> 2. We analyze the design of MSE loss from checkpoint collection process and learning objective, we also carry out weight space analysis by measuring collected weights' average L2 distance and p-value for Hartigan’s Dip Test.
>
> Looking forward to your reply!

---

### Decision · Program_Chairs · 2025-09-17

**Decision:**

Accept (poster)

**Comment:**

This paper proposes a framework named darg-and-drop LLMs to efficiently generate task-specific LoRA adapter weights based on downstream task prompts, avoiding the overhead of gradient-based optimization for LoRA adapters. Experiments on multiple adaptation tasks demonstrate the effectiveness and generalization ability of the method. The proposed method is novel and the zero-shot drag-and-drop manner for efficient adaptation is interesting and may inspire future research. All reviewers acknowledge that their questions and concerns have been addressed and favor the acceptance of this paper. The AC agrees with the reviewer comments and recommends accept for this paper.